# CLASS IMBALANCE IN FEW-SHOT LEARNING

## ABSTRACT

Few-shot learning aims to train models on a limited number of labeled samples from a support set in order to generalize to unseen samples from a query set. In the standard setup, the support set contains an equal amount of data points for each class. This assumption overlooks many practical considerations arising from the dynamic nature of the real world, such as class-imbalance. In this paper, we present a detailed study of few-shot class-imbalance along three axes: dataset vs. support set imbalance, effect of different imbalance distributions (linear, step, random), and effect of rebalancing techniques. We extensively compare over 10 state-of-the-art few-shot learning methods using backbones of different depths on multiple datasets. Our analysis reveals that 1) compared to the balanced task, the performances of their class-imbalance counterparts always drop, by up to $18.0\%$ for optimization-based methods, although feature-transfer and metric-based methods generally suffer less, 2) strategies used to mitigate imbalance in supervised learning can be adapted to the few-shot case resulting in better performances, 3) the effects of imbalance at the dataset level are less significant than the effects at the support set level. The code to reproduce the experiments is released under an open-source license.

## 1 INTRODUCTION

Deep learning methods are well known for their state-of-the-art performances on a variety of tasks (LeCun et al., 2015; Russakovsky et al., 2015; Schmidhuber, 2015). However, they often require to be trained on large labeled datasets to acquire robust and generalizable features. Few-Shot Learning (FSL) (Chen et al., 2019; Wang et al., 2019b; Bendre et al., 2020) aims at reducing this burden by defining a distribution over *tasks*, with each task containing a few labeled data points (*support set*) and a set of target data (*query set*) belonging to the same set of classes. A common way to train FSL methods is through *episodic meta-training* (Vinyals et al., 2017) with the model repeatedly exposed to batches of tasks sampled from a task-distribution and then tested on a different but similar distribution in the meta-testing phase. The prefix *"meta"* is commonly used to distinguish the high-level training and evaluation routines of meta-learning (outer loop), from the training and evaluation routines at the single-task level (inner loop).

**Limitations.** Standard meta-training overlooks many challenges stemming from real-world dynamics, such as class-imbalance (CI). The standard setting assumes that all classes in the support set contain the same number of data points, whereas in many practical applications, the number of samples for each class may vary (Buda et al., 2018; Leevy et al., 2018). Given the limited amount of data used in FSL, a small difference in the number of samples between classes could already introduce significant levels of imbalance. Most FSL methods are not designed to cope with these more challenging settings. Figure 1 exemplifies these considerations by showing that several state-of-the-art FSL methods underperform when tested under three CI regimes (linear, step, random).

**Previous work.** Previous work mainly focuses on the single imbalance case or grouping several settings into one task, offering limited insights into the effects of CI on FSL and making it challenging to quantify its effects (Guan et al., 2020; Triantafillou et al., 2020; Lee et al., 2019; Chen et al., 2020). A common approach to mitigate imbalance is Random-Shot meta-training (Triantafillou et al., 2020), which exposes the model to imbalanced tasks during meta-training. However, previous work provides little insight into the effectiveness of this procedure on the imbalanced FSL evaluation task. Furthermore, minimal work exists that investigates meta-training outcomes under an imbalanced distribution of classes at the (meta-)dataset level, while this case is common in recent FSL applications (Ochal et al., 2020; Guan et al., 2020) and meta-learning benchmarks (Triantafillou et al., 2020). The CI problem is well-known within the supervised learning community,

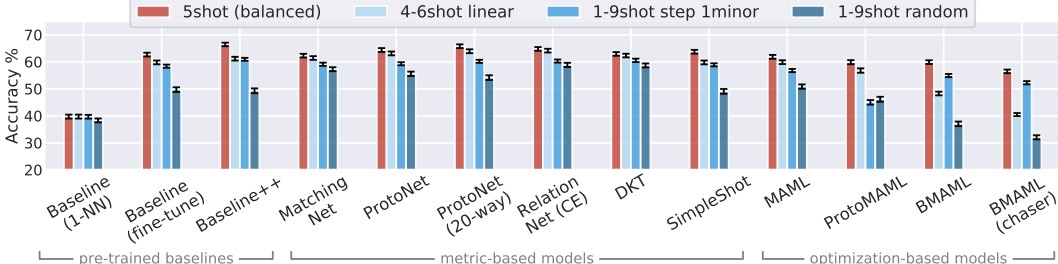

Figure 1: Accuracy (mean percentage on 3 runs) and 95% confidence intervals on FSL methods with balanced tasks (red bars) vs 3 imbalanced task (blue bars). Most methods perform significantly worse on the imbalanced tasks, as showed by the lower accuracy of the blue bars.

which has systematically produced strategies to deal with the problem, such as the popular Random Over-Sampling (Japkowicz & Stephen, 2002) that aims at rebalancing minority classes by uniform sampling. While such strategies have been extensively studied on many supervised learning problems, there is little understanding of how they behave with the recently proposed FSL methods in the low-data regime.

**Our work and main contributions.** In this paper, we provide, for the first time, a detailed analysis of the CI problem within the FSL framework. Our results show that even small CI levels can introduce a significant performance drop for all the methods considered. Moreover, we find that only a few models benefit from Random-Shot meta-training (Triantafillou et al., 2020; Lee et al., 2019; Chen et al., 2020) over the classical (balanced) episodic meta-training (Vinyals et al., 2017); while pairing the meta-training procedures with Random Over-Sampling offers a substantial advantage. The experimental results show that imbalance severity at the dataset level depends on the size of the dataset. Our *contributions* can be summarized as follows:

1. A systematic, comprehensive and in-depth study of the effects of CI within the FSL framework along three axes: (i) dataset vs. support set imbalance, (ii) effect of different imbalance distributions (linear, step, random), (iii) effect of rebalancing techniques, such as random over-sampling and the recently proposed Random-Shot meta-training (Triantafillou et al., 2020).

2. We reveal novel insights into the meta-learning and support set adaptation capabilities to the CI regime, supported by extensive results on over 10 FSL methods with different imbalance settings, backbones, support set sizes, and datasets.

3. We provide insight into the previously unaddressed CI problem in the (meta-)training dataset, showing that the effects of imbalance at the dataset level are less significant than the effects at the support set level.

## 2 RELATED WORK

### 2.1 CLASS IMBALANCE

In classification, imbalance occurs when at least one class (the majority class) contains a higher number of samples than the others. The classes with the lowest number of samples are called minority classes. If uncorrected, conventional supervised loss functions, such as (multi-class) cross-entropy, skew the learning process in favor of the majority class, introducing bias and poor generalization toward the minority class samples (Buda et al., 2018; Leevy et al., 2018). Imbalance approaches are categorized into three groups: data-level, algorithm-level, and hybrid. *Data-level* strategies manipulate and create new data points to equalize data sampling. Popular data-level methods include Random Over-Sampling (ROS) and Random Under-Sampling (RUS) (Japkowicz & Stephen, 2002). ROS randomly resamples data points from the minority classes, while RUS randomly leaves out a randomly selected portion of the majority classes to decrease imbalance levels. *Algorithm-level* strategies use regularization or minimization of loss/cost functions. Weighted loss is a common approach where each sample's loss is weighted by the inverse frequency of that sample's class. Focal loss (Lin et al., 2017) is another type of cost function that has seen wide success. *Hybrid* methods combine one or more types of strategies (e.g. Two-Phase Training, Havaei et al. (2017)).

**Modeling Imbalance.** The object recognition community studies class imbalance using real-world datasets or distributions that approximate real-world imbalance (Buda et al., 2018; Johnson & Khoshgoftaar, 2019; Liu et al., 2019). Buda et al. (2018) note that two distributions can be used: *linear* and *step* imbalance (defined in our methodology Section 3). At large-scale, datasets with many samples and classes tend to follow a *long-tail* distribution (Liu et al., 2019; Salakhutdinov et al., 2011; Reed, 2001), with most of the classes occurring with small frequency and a few classes occurring with high frequency. Our work primarily focuses on the *tail-end* of the distribution and does not consider the case of large sample size. Therefore, we do not examine the long-tail mechanisms.

## 2.2 FEW-SHOT LEARNING

FSL methods can be broadly categorized into metric-learning, optimization-based, hallucination, data-adaptation, and probabilistic approaches (Chen et al., 2019). *Metric-learning* approaches such as Prototypical Networks (Snell et al., 2017), Relation Networks (Sung et al., 2017), the Neural Statistician (Edwards & Storkey, 2017) and Matching Networks (Vinyals et al., 2017), learn a feature extractor capable of parameterizing images into embeddings, and then use distance metrics to classify mapped query samples based on their distance to support points. *Optimization-based* approaches such as MAML (Finn et al., 2017) and Meta-Learner LSTM (Ravi & Larochelle, 2016)), are meta-trained to use guided optimization steps on the support set for quick adaptation. *Hallucination* or *data augmentation* techniques perform affine and color transformations on the support set to create additional data points (Zhang et al., 2018). *Probabilistic* methods use Bayesian inference to learn and classify samples, for example, the recently proposed Deep Kernel Transfer (DKT) (Patacchiola et al., 2020), which uses Gaussian Process at inference time. We use the term *domain adaptation* to represent those approaches using standard *transfer-learning* with a pre-training stage on a large set of classes and a fine-tune stage on the support set – examples are Baseline and Baseline++ from Chen et al. (2019), and the recently proposed Transductive Fine-Tuning from Dhillon et al. (2020). The details of the methods used in our experiments are reported in Appendix A. For completeness, it is worth mentioning *Incremental Few-Shot Learning* (Ren et al., 2018; Gidaris & Komodakis, 2018; Hariharan & Girshick, 2017) which is an extension of FSL. It considers maintaining performance on base classes (meta-training dataset) while incrementally learning about novel classes using limited data, typically without re-training from scratch on all data. Here, we focus on studying how imbalance affects the learning of novel classes only; therefore, we will not consider incremental FSL further.

## 2.3 IMBALANCE IN FEW-SHOT AND META LEARNING

Class Imbalance in the low-data regime has received some attention, although the current work is not comprehensive (Guan et al., 2020; Triantafillou et al., 2020; Lee et al., 2019; Chen et al., 2020). We identify that in FSL, *class-imbalance* occurs at two levels: the task-level and the meta-dataset level. At the task level, class-imbalance occurs in the support set or the query-set, directly affecting learning and evaluation procedures. Class imbalance at the meta-dataset level is caused by imbalanced dataset classes in one (or more) of the three data splits: meta-training, meta-validation, meta-testing. This disproportion affects the distribution of tasks that a model is exposed to during meta-training, affecting their ability to generalize to new tasks. In Figure 2, we highlight the differences between imbalanced task and imbalanced meta-dataset. Related to, but distinct from, these two class-imbalance types is *task-distribution* imbalance (Lee et al., 2019); skewed task-distribution can occur as a result of meta-dataset level class-imbalance or as a result of the task-sampling procedure. In extreme cases, task-distribution imbalance can lead to out-of-distribution tasks during meta-evaluation. Task-distribution imbalance has already received some attention (Lee et al., 2019; Cao et al., 2020); therefore, it will not be considered in this work.

**Class Imbalance in Tasks.** Triantafillou et al. (2020) uses imbalanced support sets to create a more realistic and challenging benchmark for meta-learning. The authors use random-shot tasks with randomly selected classes (way) and samples (shot), which replace the balanced task during meta-training and meta-evaluation. A similar idea is explored by Lee et al. (2019), Chen et al. (2020), and Guan et al. (2020) with the last two using a fixed number of classes (way). However, none of these works quantify the impact of class-imbalance on the FSL task nor the advantages of Random-Shot meta-training. Lee et al. (2019) explores a small range of class-imbalance in the support set. However, details into the effects of class-imbalance are lost when combined with task-distribution imbalance, making it challenging to attribute any performance changes caused by class-imbalance. Chen et al. (2020) explore a pure class-imbalance problem on the support set, but their

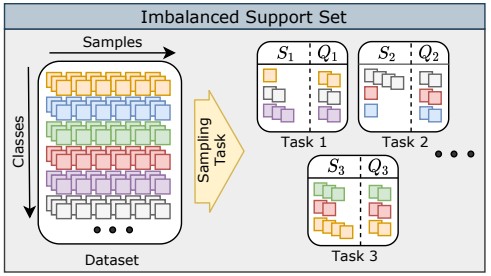 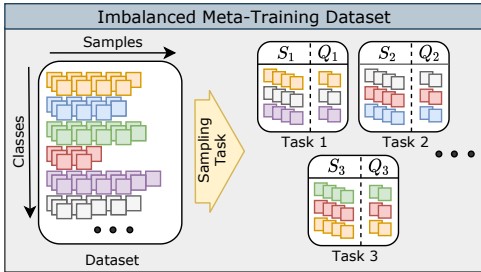

Figure 2: The two types of imbalance settings investigated in this work. **Left:** Imbalanced support set. Classes are balanced at the dataset level, but tasks are imbalanced by one of $\mathcal{I}$-distributions: *linear* (task 1), *step* (task 2), and *random* (task 3) **Right:** Imbalanced meta-training dataset. Classes are imbalanced at the dataset level, but all the support sets are *balanced* at the task level. Following standard practice in the literature, query sets are kept balanced in both settings.

analysis is limited to just two methods (their proposal and MAML). In Guan et al. (2020), meta-learning is applied on aerial imagery, exploring step imbalance ranging from 5 to 140 samples per class (shot); however, only two FSL methods are compared (Prototypical Networks and their RF-MML method). Previous work provides limited insight into class-imbalance at the task level.

**Class Imbalance in the Meta-Dataset.** Standard meta-datasets (e.g. Mini-ImageNet) can be swapped for other domain-specific datasets, such as CUB (Wah et al., 2011), VGG Flowers (Nils-back & Zisserman, 2008), and others (Triantafillou et al., 2020). These datasets sometimes contain an unequal number of class samples, but previous work has never reported the effects of class-imbalance in the meta-training dataset (Guan et al., 2020; Triantafillou et al., 2020; Lee et al., 2019; Chen et al., 2020). We emphasize that studying the impact of imbalance at this level is important since imbalanced domain-specific meta-datasets are common in real-world applications (Guan et al., 2020; Ochal et al., 2020) and recent benchmarks (Triantafillou et al., 2020). Our work is the first to provide quantitative insights into this setting.

## 3 METHODOLOGY

### 3.1 STANDARD FSL

A standard $K$-shot $N$-way FSL classification task is defined by a small *support set*, $\mathcal{S} = \{(x_1, y_1), ..., (x_s, y_s)\} \sim \mathcal{D}$, containing $N \times K$ image-label pairs drawn from $N$ unique classes with $K$ samples per class ($|\mathcal{S}| = K \times N$). The goal is to correctly predict labels for a *query set*, $\mathcal{Q} = \{(x_1, y_1), ..., (x_t, y_t)\} \sim \mathcal{D}$, containing a different set of $M$ samples drawn from the same $N$ classes (i.e. $\mathcal{Q}^{(x)} \cap \mathcal{S}^{(x)} = \emptyset$ and $\mathcal{Q}^{(y)} \equiv \mathcal{S}^{(y)}$). The support set can also be referred to as *sample set* and the query set as *target set*.

### 3.2 CLASS-IMBALANCED FSL

We define a class-imbalanced FSL task as a $K_{min}$-$K_{max}$-shot $N$-way $\mathcal{I}$-distribution task. Similarly to the standard FSL task, a model is given a small *support set*, $\mathcal{S} \sim \mathcal{D}$ and a *query set*, $\mathcal{Q} \sim \mathcal{D}$, containing a different set of samples drawn from the same $N$ classes. However, in the imbalance case, the support set contains between $K_{min}$ to $K_{max}$ (inclusive) number of samples per class distributed according to the imbalance $\mathcal{I}$-distribution, where $\mathcal{I} \in \{linear, step, random\}$ (Buda et al., 2018). Similarly, the query set can contain $M_{min}$ to $M_{max}$ samples per class distributed according to the $\mathcal{I}$-distribution. In our experiments, we keep a balanced query set ($M = M_{min} = M_{max}$) for fair evaluation.[1] For brevity, but without loss of generality, we define imbalance $\mathcal{I}$-distribution in relation to the support set (see Figure 2) as:

- *Linear imbalance*. The number of class samples, $K_i$, for classes $i \in \{1..N\}$ is defined by:

$$K_i = \texttt{round}\left(K_{min} - c + (i-1) \times (K_{max} + c * 2 - K_{min})/(N-1)\right), \quad (1)$$

---

[1]This is a standard procedure used in the class-imbalance literature (Buda et al., 2018), which reduces the number of variables and allows isolating the effect of imbalance. Note that an imbalanced query set would influence methods such as SCA (Antoniou & Storkey, 2019), which use the query set as an additional unlabeled set during the inner-loop. We do not consider such methods in our experiments since they assume immediate access to the query set, which limits their practical application.

where $c = 0.499$ for rounding purposes. For example, this means that for linear 1-9-shot 5-way task, $K_i \in \{1, 3, 5, 7, 9\}$, and for linear 4-6-shot 5-way task $K_i \in \{4, 4, 5, 6, 6\}$.

- *Step imbalance*. The number of class samples, $K_i$, is determined by an additional variable $N_{min}$ specifying the number of minority classes. Specifically, for classes $i \in \{1..N\}$:

$$K_i = \begin{cases} K_{min}, & \text{if } i \leq N_{min}, \\ K_{max}, & \text{otherwise.} \end{cases} \quad (2)$$

For example, in a step 1-9-shot 5-way task with 1 minority class $K_i \in \{1, 9, 9, 9, 9\}$.

- *Random imbalance*. The number of class samples, $K_i$, is sampled from a uniform distribution, i.e. $K_i \sim \texttt{Unif}(K_{min}, K_{max})$, with $K_{min}$ and $K_{max}$ inclusive. This is appropriate for the problem at hand (small number of classes), but it could be replaced by a Zipf/Power Law (Reed, 2001) for a more appropriate imbalance in problems with a large number of classes.

We also report the imbalance ratio $\rho$, which is a scalar identifying the level of class-imbalance; this is often reported in the CI literature for the supervised case (Buda et al., 2018). We define $\rho$ to be the ratio between the number of samples in the majority and minority classes in the support set:

$$\rho = \frac{K_{max}}{K_{min}}. \quad (3)$$

### 3.3 CLASS-IMBALANCED META-DATASET

Training FSL methods involves three phases: meta-training, meta-validation, and meta-testing. Each phase samples tasks from a different dataset, $\mathcal{D}_{train}$, $\mathcal{D}_{val}$, and $\mathcal{D}_{test}$, respectively. A balanced dataset contains $\mathcal{D}_*^N$ classes with $\mathcal{D}_*^K$ samples per class, where $* \in \{train, val, test\}$. However, in the real-world, datasets can contain any number of samples with imbalance. For fair evaluation, we control dataset imbalance according to the $\mathcal{I}$-distribution described in Section 3.2 but with $K_{min}$, $K_{max}$, $N$, $N_{min}$ changed for $\mathcal{D}_*^{K_{min}}$, $\mathcal{D}_*^{K_{max}}$, $\mathcal{D}_*^N$, $\mathcal{D}_*^{N_{min}}$. Similarly, we report the imbalance ratio $\rho$. In our experiments, we apply imbalance only at the meta-training stage to limit the factors of interest, but a similar procedure could be used at the meta-testing and meta-validation stages.

### 3.4 REBALANCING TECHNIQUES AND STRATEGIES

**Random Over-Sampling.** We apply *Random Over-Sampling (ROS)* and for each class-imbalanced task, we match the number of support samples in the non-majority classes to the number of support samples in the majority class, $K_i = \max_i(K_i)$. This means that for $\mathcal{I} \in \{linear, step\}$, the number of samples in each class is equal to $K_{max}$. We match $K_i$ to $\max_i(K_i)$ by resampling uniformly at random the remaining $\max_i(K_i) - K_i$ support samples belonging to class $i$, and then appending them to the support set. When applying ROS with augmentation (*ROS+*), we perform further data transformation on the resampled supports. A visual representation of a class imbalanced task after applying ROS and ROS+ is presented in the Appendix A (Figure 7).

**Random-Shot Meta-Training.** We apply *Random-Shot* meta-training similarly to the *Standard* episodic (meta-)training (Vinyals et al., 2017) but with the balanced tasks exchanged with $K_{min}$-$K_{max}$-shot *random*-distribution tasks, as defined above. We use random-distribution following previous work (Triantafillou et al., 2020; Lee et al., 2019), since in real-world applications, the actual imbalance distribution is likely to be unknown at (meta-)evaluation time.

**Rebalancing Loss Functions.** We apply two rebalancing loss functions: *Weighted Loss* (Buda et al., 2018) and *Focal Loss* (Lin et al., 2017). Both of them have been applied to the inner loop of optimization-based methods. Full details are reported in the supplementary material (Appendix A).

## 4 EXPERIMENTS

### 4.1 SETUP

**Class Imbalance Scenarios and Tasks.** We address two class-imbalance scenarios within the FSL framework: 1) imbalanced support set, and 2) imbalanced meta-training dataset. For the imbalanced support set scenario, we first focus on the very low-data range with an average support set size of 25 samples (5 avr. shot). We train FSL models using *Standard* (episodic) meta-training (Vinyals et al., 2017) using 5-shot 5-way tasks, as well as *Random-Shot* meta-training (Triantafillou et al., 2020;

Lee et al., 2019; Chen et al., 2020) using 1-9shot 5-way random-distribution tasks (as described in Section 3.2). We pre-train baselines (i.e., Fine-Tune, 1-NN, Baseline++) using mini-batch gradient descent, and then fine-tune on the support or perform a 1-NN classification. We evaluate all baselines and models using a wide range of imbalanced meta-testing tasks. In contrast to previous work, we evaluate models using two additional imbalance distributions, *linear* and *step*; this allows us to control the imbalance level deterministically and provide insights from multiple angles. For the imbalanced meta-dataset scenario, we vary the class distributions of the meta-training datasets. We isolate this level of imbalance by meta-training and meta-evaluating on balanced FSL tasks. All main experiments are repeated three times with different initialization seeds. Each data point represents the average performance of over 600 meta-testing tasks per run.

**Additional details.** We adapted a range of 11 unique baselines and FSL methods: Fine-tune baseline (Pan & Yang, 2010), 1-NN baseline, Baseline++ (Chen et al., 2019), SimpleShot (Wang et al., 2019a), Prototypical Networks (Snell et al., 2017), Matching Networks (Vinyals et al., 2017), Relation Networks (Sung et al., 2017), MAML (Finn et al., 2017), ProtoMAML (Triantafillou et al., 2020), DKT (Patacchiola et al., 2020), and Bayesian MAML (BMAML) (Yoon et al., 2018). Implementation details of these algorithms are supplied in Appendix A. We used a 4 layer convolutional network as backbone for each model, following common practice (Chen et al., 2019). We train and evaluate all methods on MiniImageNet (Ravi & Larochelle, 2016; Vinyals et al., 2017), containing 64 classes with 600 image samples each. In the imbalanced meta-dataset setting, we half the Mini-ImageNet dataset to contain 300 samples per class on average, and control imbalance as described in section 3.3. For full implementation details, see Appendix A.

## 4.2 CLASS IMBALANCED SUPPORT SET

**Effect of Class Imbalance with Standard Meta-Training.** Figure 1 highlights the crux of the class-imbalance problem at the support set level. Specifically, the figure shows *standard* meta-trained FSL models (Vinyals et al., 2017) and pre-trained baselines, evaluated on the balanced 5-shot 5-way task and three imbalanced tasks. We observe that introducing even a small level of imbalance (linear 4-6-shot 5-way, $\rho = 1.5$) produces a significant[2] performance difference for 6 out of 13 algorithms, compared with the balanced 5-shot task. The average accuracy drop is $-1.5\%$ for metric-based models and $-8.2\%$ for optimization-based models. On tasks with a larger imbalance (1-9shot random, $\rho = 9.0$), the performance drops by an average $-8.4\%$ for metric-based models and $-18.0\%$ for optimization-based models compared to the balanced task. Interestingly, despite the additional 12 samples in the support set in 1-9shot step tasks with 1 minority class ($\rho = 9.0$), the performance drops by $-5.0\%$ on the balanced task with 25 support samples in total.

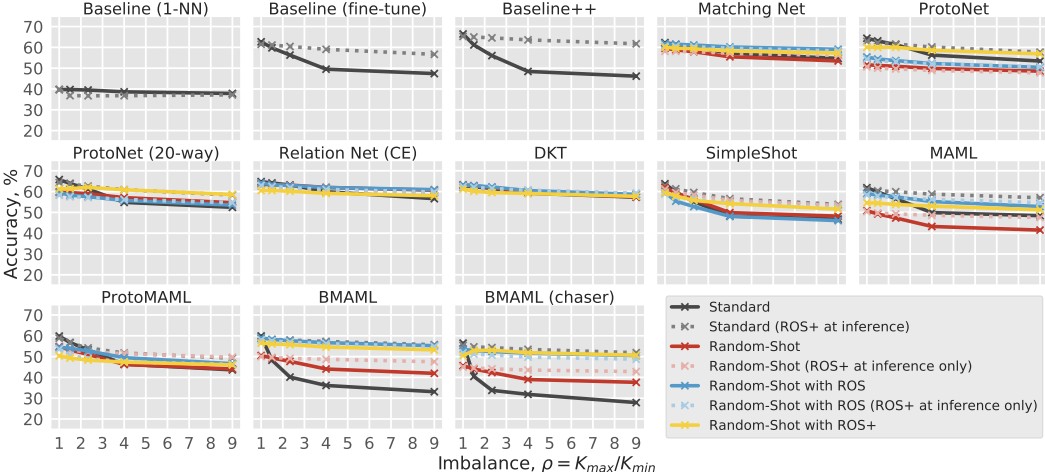

Figure 3: Standard episodic training (Vinyals et al., 2017) vs. random-shot episodic training (Triantafillou et al., 2020). We explore pairing methods with Random Over-Sampling (ROS) without and with augmentation (ROS+).

---

[2]Non-overlapping 95% confidence intervals indicate 'significant' performance difference.

**Standard vs. Random-Shot Meta-Training.** In Figure 3, we show the accuracy for increasing imbalance levels ($\rho$) using evaluation tasks with a linear distribution and fixed support set size ($K_i \approx$ 5) for a fair comparison. Comparing *Standard* and *Random-Shot* meta-training (solid black and solid red lines) reveals that only a few methods benefit from Random-Shot meta-training. On the balanced 5-shot task, we observe a $-6.0\%$ decrease in accuracy, caused by Random-Shot over Standard meta-training. On the imbalanced 1-9shot random task, Random-Shot offers a limited improvement over the Standard, with a significant increase in performance for only 3 out of 10 models. Those improvements include $+2.5\%$ for Relation Net, and $+6.6\%$ for BMAMLs. These results suggest that exposing FSL methods to imbalanced tasks during meta-training does not automatically lead to improved performance at meta-test time. Interestingly, in an extreme imbalance case (1-21shot step 4minor, Appendix E) only ProtoNet and RelationNet obtained a significantly higher performance with Random-Shot ($+18\%$ compared to Standard). This suggests that the advantage may emerge from coupling Random Shot with the prototype calculation mechanism unique to those methods. The results also suggest that some models have natural robustness to imbalance: Relation Net, MatchingNet, and DKT only drop slightly compared to other methods.

**Random-Shot with Random Over-Sampling.** In Figure 3, we observe that the performances of optimization-based methods such as MAML and BMAML significantly improve by applying random over-sampling with augmentation (ROS+) and without augmentation (ROS). In the largest imbalance case in the graph ($\rho$=9), we observe that models using ROS+ at inference (dotted and yellow lines) improve over the Standard (solid black) by $+6.7\%$; in particular, optimization-based methods improve by $+12.2\%$, fine-tune baselines by $+7.4\%$, and metric-based by $+2.8\%$. In the imbalanced task, the least affected model is MatchingNet only dropping $-1.9\%$ compared to the balanced task; we provide a list of top-50 performing models in Table 3 (Appendix C.1). Standard (ROS+ at inference) achieves the highest average performance gains ($+8.5\%$); tieing for second best is Random-Shot with ROS (ROS+ at inference) with $+6.9\%$ and Random-Shot with ROS+ ($+6.4\%$). We breakdown the results by type in Appendix C.2 (Figure 9).

**Imbalance with More Shots.** We explored additional settings with a higher number of shots, see Figure 4. Specifically, we train models using Random-Shot meta-training with 1-29 shot and 1-49 random episodes. We then evaluate those models on imbalanced tasks with an average number of 15 shots and 25 shots, respectively. The bottom row of Figure 4 shows the difference in performance between the imbalance and balanced tasks. We observe that for the high-shot condition (right column), the general model performance increases while the models are less affected by imbalance; however, the gap with respect to the balanced condition remains significant. Models achieve 55-60% of their performance on the balanced task within first 5 avr. shots; increasing the number of shots to 15 only boosts their performance by $+7\%$. This may explain why imbalance will have an inevitable impact on small classification tasks: better performance achieved via a higher numbers of support samples in the majority classes, does not offset the performance lost due to lack of samples in the minority classes. In Appendix C.2, we breakdown the results for each model type.

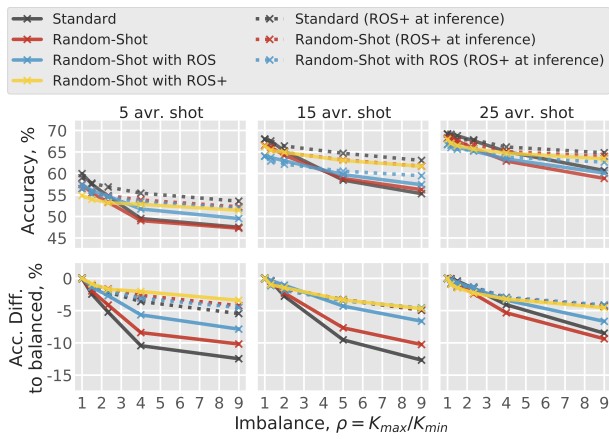

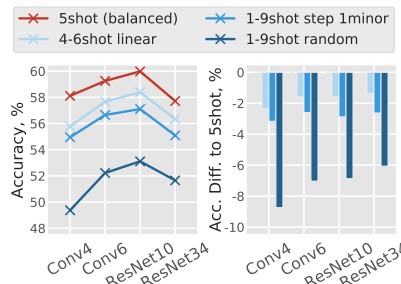

Figure 5: Combined average model performance against different backbones and imbalanced tasks. *Left:* combined performance of all models and training scenarios. *Right:* relative performance w.r.t. the balanced task.

Figure 4: Comparing imbalance levels via support sets of different size. Each line represents the average across all models in each training and imbalance setting.

**Backbones.** In Figure 5, we report the combined average accuracy of all models and imbalance strategies against different backbones (Conv4, Conv6, ResNet10, ResNet34). Overall, deeper backbones seem to perform slightly better on the imbalanced tasks, suggesting a higher tolerance for imbalance. For instance, using Conv4 gave $-8.6\%$ difference between the balanced and the 1-9shot random task, while using ResNet10 the gap is smaller ($-6.8\%$). The performance degradation observed with ResNet34 is similar to that reported by Chen et al. (2019), and is most likely caused by the intrinsic instability of meta-training routines on larger backbones. In Appendix C.3, we breakdown the results across different models and training strategies.

**Precision and Recall.** Looking at the precision and recall tables in Appendix C.4, provides additional insights about each algorithm. For instance, DKT (Patacchiola et al., 2020) shows very strong performance in classes with a small number of shots and well-balanced performances for higher shots. This may be due to the partitioned Bayesian one-vs-rest scheme used for classification by DKT, with a separate Gaussian Process for each class; this could be more robust to imbalance. BMAML, on the other hand, fails to correctly classify samples with $K = 1$ and $K = 3$ samples, showing that the method has a strong bias towards the majority classes.

**Rebalancing Cost Functions** We applied Focal Loss (Lin et al., 2017) and Weighted Loss (Buda et al., 2018; Japkowicz & Stephen, 2002) to the inner-loop of optimization-based methods at inference time only. Results in Figure 6 and Appendix D.1 show that overall, Focal Loss is not as effective as ROS+ techniques. However, ROS+ and Weighted Loss perform very similarly, suggesting a similar effect on the imbalanced task. The advantage of using ROS/ROS+ is their versatility; any FSL algorithm can use ROS, while algorithm-level balancing approaches, such as Weighted Loss, do not straightforwardly extend to metric-learning methods.

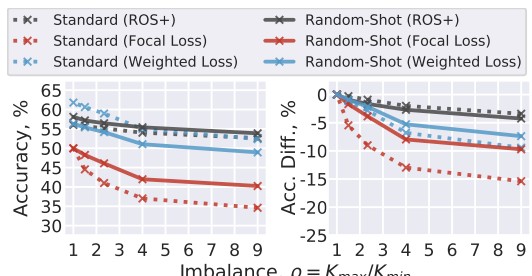

Figure 6: Combined average model performance against re-balancing strategies applied at test-time only. *Left:* all models and training scenarios. *Right:* performance w.r.t. the balanced task.

### 4.3 CLASS IMBALANCED META-DATASET

**Imbalanced Mini-ImageNet.** To induce dataset imbalance, we half the number of samples per class in Mini-ImageNet. In table 1 (left), we show the accuracy after training via standard episodic (meta-)training (Vinyals et al., 2017) with (balanced) 5-shot 5-way tasks. In this particular scenario, we use significantly higher imbalance levels ($\rho = 19$) compared to those in the previous section; despite this, we observe small, insignificant performance differences between balanced and imbalanced conditions. In additional experiments, we further reduced the dataset size to contain a total of 4800 images and 32 randomly selected classes. In Figure 10 (Appendix C.5), we observe a more significant performance drop as we increase the number of minority classes. Meta-evaluating on CUB showed a similar trend, with an average drop of $-1.6\%$ on the most extreme imbalance setting: 30-510 step with 24 minority classes and $\rho = 17.0$ (Appendix C.5). When we breakdown the results by model in Appendix C.5, we observe that optimization-based approaches and fine-tune baselines have a slight advantage over the metric-based, most likely due to the ability to adapt during inference. Interestingly, in this setting RelationNet performs the worse with a drop of $-4.3\%$ w. r. t. the balanced task in the most extreme setting (24 minority, Mini-ImageNet).

**Additional results.** To evaluate the performance under a strong dataset shift, we evaluated Mini-ImageNet trained models on tasks sampled from CUB-200-2011. In Table 1 (right), we observe that models are not affected at all by the imbalanced setting despite the harder scenario. In Appendix D, we provide additional experiments with BTAML (Lee et al., 2019), and an analysis of the correlation between meta-dataset size and performance

## 5 DISCUSSION

**FSL robustness to class imbalance.** All examined FSL methods are susceptible to class imbalance, although some show more robustness (e.g., Matching Net, Relation Net, and DKT).

Table 1: Training on **imbalanced meta-training dataset**. Imbalanced distributions represent $\rho = 19$ ($\mathcal{D}^{K_{min}} = 30$, $\mathcal{D}^{K_{max}} = 570$) with $step$ imbalance containing $\mathcal{D}^{N_{min}} = 32$ minority classes (out of 64 available in the dataset). Small differences in accuracy between $balanced$ and $\mathcal{I}$-distributions, suggest insignificant effect of imbalance at dataset level. **Left:** Evaluation on the meta-testing dataset of Mini-ImageNet. **Right:** Evaluation on the meta-testing dataset of CUB.

| Imbalance $\mathcal{I}$ | Imbalanced Mini-ImageNet | | | | Imbalanced Mini-ImageNet $\rightarrow$ CUB | | | |
|---|---|---|---|---|---|---|---|---|
| | balanced | linear | random | step | balanced | linear | random | step |
| Baseline (1-NN) | $42.69_{\pm0.66}$ | $43.42_{\pm0.68}$ | $42.15_{\pm0.66}$ | $41.45_{\pm0.65}$ | $43.21_{\pm0.68}$ | $43.42_{\pm0.69}$ | $43.39_{\pm0.69}$ | $42.19_{\pm0.66}$ |
| Baseline (fine-tune) | $51.26_{\pm0.70}$ | $50.13_{\pm0.69}$ | $54.16_{\pm0.72}$ | $52.47_{\pm0.70}$ | $53.19_{\pm0.71}$ | $51.95_{\pm0.72}$ | $53.52_{\pm0.72}$ | $52.68_{\pm0.70}$ |
| Baseline++ | $48.44_{\pm0.65}$ | $47.18_{\pm0.64}$ | $51.47_{\pm0.67}$ | $51.88_{\pm0.69}$ | $49.38_{\pm0.69}$ | $46.83_{\pm0.67}$ | $50.48_{\pm0.67}$ | $48.42_{\pm0.67}$ |
| Matching Net | $58.26_{\pm0.68}$ | $58.24_{\pm0.69}$ | $58.45_{\pm0.68}$ | $56.53_{\pm0.69}$ | $50.92_{\pm0.74}$ | $51.32_{\pm0.73}$ | $50.77_{\pm0.76}$ | $50.51_{\pm0.73}$ |
| ProtoNet | $60.65_{\pm0.70}$ | $59.17_{\pm0.68}$ | $60.16_{\pm0.70}$ | $58.69_{\pm0.72}$ | $52.86_{\pm0.73}$ | $51.85_{\pm0.72}$ | $52.06_{\pm0.71}$ | $52.42_{\pm0.71}$ |
| ProtoNet (20-way) | $60.91_{\pm0.70}$ | $60.64_{\pm0.70}$ | $60.37_{\pm0.70}$ | $58.83_{\pm0.70}$ | $52.80_{\pm0.72}$ | $52.60_{\pm0.74}$ | $52.31_{\pm0.73}$ | $51.33_{\pm0.72}$ |
| Relation Net (CE) | $\mathbf{62.78}_{\pm0.70}$ | $\mathbf{61.39}_{\pm0.69}$ | $\mathbf{62.35}_{\pm0.70}$ | $57.93_{\pm0.72}$ | $54.32_{\pm0.68}$ | $\mathbf{54.41}_{\pm0.71}$ | $52.13_{\pm0.65}$ | $49.90_{\pm0.62}$ |
| DKT | $58.09_{\pm0.69}$ | $57.59_{\pm0.68}$ | $57.81_{\pm0.67}$ | $55.91_{\pm0.67}$ | $\mathbf{54.62}_{\pm0.71}$ | $54.19_{\pm0.71}$ | $54.86_{\pm0.72}$ | $\mathbf{54.44}_{\pm0.71}$ |
| SimpleShot | $59.55_{\pm0.72}$ | $59.78_{\pm0.71}$ | $58.74_{\pm0.72}$ | $\mathbf{58.89}_{\pm0.71}$ | $53.16_{\pm0.71}$ | $53.46_{\pm0.71}$ | $52.88_{\pm0.72}$ | $52.87_{\pm0.71}$ |
| MAML | $54.43_{\pm0.69}$ | $55.14_{\pm0.72}$ | $54.97_{\pm0.73}$ | $54.30_{\pm0.70}$ | $53.46_{\pm0.67}$ | $53.26_{\pm0.70}$ | $\mathbf{55.14}_{\pm0.67}$ | $53.96_{\pm0.69}$ |
| ProtoMAML | $51.31_{\pm0.72}$ | $54.57_{\pm0.69}$ | $45.94_{\pm0.73}$ | $53.56_{\pm0.71}$ | $48.52_{\pm0.72}$ | $51.25_{\pm0.69}$ | $45.27_{\pm0.70}$ | $51.64_{\pm0.67}$ |
| Avr. Diff. to balanced | 0.0 | -0.1 | -0.2 | -0.7 | 0.0 | -0.2 | -0.3 | -0.6 |

Optimization-based methods and fine-tune baselines suffer more as they use conventional supervised loss functions in the inner-loop which are known to be particularly susceptible to imbalance (Buda et al., 2018; Johnson & Khoshgoftaar, 2019; Japkowicz & Stephen, 2002). Moreover, the problem of class imbalance persists as the backbone complexity and support set size increase. Those results suggest that current solutions will offer sub-optimal performance in real-world few-shot problems.

**Effectiveness of Random-Shot meta-training.** Our experiments test a simple solution to class imbalance that has been popular in the meta-learning community – Random-Shot meta-training (Triantafillou et al., 2017; Lee et al., 2019; Chen et al., 2020). Contrarily to popular belief, our findings reveal that this method is scarcely effective when applied by itself. Extensive analysis and validation performance through epochs (see Appendix B) suggest that these results are genuine and unlikely to be the result of inappropriate parameter tuning. This finding has an important consequence, suggesting that robustness to imbalance cannot be obtained by the simple exposure to imbalanced tasks.

**Effectiveness of re-balancing procedures.** The results suggest that a simple procedure, Random Over-Sampling (ROS), is quite effective in tackling class imbalance issues. Therefore, we encourage the community to include it in their evaluation as ROS is simple to implement, and it can be applied to almost any algorithm. However, ROS does not provide any particular advantage to methods in the highest performance ranking levels, like MatchingNet and DKL. This could be due to diminishing returns and should be investigated on a case-by-case basis.

**Effect of imbalance at the meta-dataset level.** Our results suggest that imbalance in the meta-dataset has minimal effect on the meta-learning procedure. This could result from standard episodic (meta-)training that samples classes with equal probability and causes natural re-sampling. Likely, datasets with lower intra-class variation and larger imbalance (Liu et al., 2019; Wang et al., 2017; Salakhutdinov et al., 2011) could produce more dramatic performance changes.

## 6 CONCLUSION

In this work, we have provided a detailed analysis of class-imbalance in FSL, showing that class-imbalance at the support set level is problematic for many methods. We found that most metric-based models present a built-in robustness to support-set imbalance, while in optimization-based models imbalance issues can be alleviated using oversampling. In our experiments, Random-Shot meta-training provided minimal benefits suggesting that meta-learning methods do not learn to balance from random-shot episodes alone. Results on meta-dataset imbalance showed just a small negative effect, but this effect is not as dramatic as with the task-level imbalance. In future work, the insights gained with our investigation could be used to design novel few-shot methods that can guarantee a stable performance under the imbalance condition.

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

# A    IMPLEMENTATION DETAILS

## A.1    DATASETS

For the imbalanced support set experiments, we used MiniImageNet (Vinyals et al., 2017; Ravi & Larochelle, 2016) following the same version popular version as (Ravi & Larochelle, 2016; Cheng et al., 2018). All meta-learning models used 64, 16, 20 classes for the meta-training $\mathcal{D}_{train}$, meta-validation $\mathcal{D}_{val}$, and meta-testing $\mathcal{D}_{test}$ datasets, respectively, with each class containing 600 samples. All images are resized to 84 by 84px. For the feature-transfer baselines (1-NN, fine-tune, and Baseline++), we used a conventionally partitioned training and validation datasets. Specifically, we combined $\mathcal{D}_{train}$ and $\mathcal{D}_{val}$ classes (i.e., $64 + 16 = 80$ classes), then partitioned the samples of each class into 80% - 20% split for pre-training - validation, forming $\mathcal{D}'_{train}$ and $\mathcal{D}'_{val}$ (where $\mathcal{D}'^{(y)}_{train} \equiv \mathcal{D}'^{(y)}_{val}$). Thus, the baseline were trained on the same number of training samples as the meta-learning methods, albeit with more classes and less samples per class. All models were evaluated on FSL tasks sampled from $\mathcal{D}_{test}$.

For the imbalanced meta-dataset experiments, we used two variants of MiniImageNet. In the first, referring to Table 13, we halved the average number of samples per class in the meta-training dataset $D_{train}$ to allow us to introduce imbalance artificially into the dataset. In the second scenario, referring to Figure 10, we reduced the size of the meta-training by a more considerable degree. Specifically, the size of the meta-training dataset was controlled to contain a total of 4800 images distributed among 32 randomly selected classes from the meta-training dataset of Mini-ImageNet. For meta-learning methods, we kept the original 16 and 20 classes for meta-validation and meta-testing with 600 samples each. To allow as fair comparison as possible, we used the same meta-training datasets for the baselines and meta-learning models. However, the baselines used a balanced validation set created from the leftover samples from the original meta-training dataset.

For the imbalanced meta-dataset experiments, we also evaluated on tasks sampled from 50 randomly selected classes from CUB-200-2011 (Wah et al., 2011), following the same line of work as Chen et al. (2019).

## A.2    TRAINING PROCEDURE.

All methods follow a similar three-phase learning procedure: meta-training, meta-validation, and meta-testing. During meta-training, an FSL model was exposed to 100k tasks sampled from $\mathcal{D}_{train}$. After every 500 tasks, the model was validated on tasks from $\mathcal{D}_{val}$ and the best performing model was updated. At the end of the meta-training phase, the best model was evaluated on tasks sampled from $\mathcal{D}_{test}$. The baselines (i.e., fine-tune, 1-NN, Baseline++) follow a similar three-phase procedure but with the meta-training / meta-validation phases exchanged for conventional pre-training / validation on mini-batches (of size 128) sampled from $\mathcal{D}'_{train}$ and $\mathcal{D}'_{val}$ as outlined above. In all three meta- phase's tasks, we used 16 query samples per class, except for the 20-way Prototypical Network, where we used 5 query samples per class during meta-training to allow for a higher number of samples in the support set.

**Meta-/Pre- Training Details.**    In the imbalanced support set setting, we meta-train FSL methods using *standard* episodic meta-training (Vinyals et al., 2017) using 5-shot 5-way tasks. We also explore *random-shot* episodic training (Lee et al., 2019) using 1-9shot 5-way random-distribution tasks (as described in section 3). We meta-/pre- trained on 100k tasks/mini-batches, using a learning rate of $10^{-3}$ for the first 50k episodes/mini-batches, and $10^{-4}$ for the second half. The baselines and SimpleShot are trained using 100k balanced mini-batches with a batch size of 128. All methods were meta-validated on 200 tasks/mini-batches every 500 meta-training tasks/mini-batches to select the best performing model.

**Meta-Testing.**    The final test performances were measured on a random sample of 600 tasks. We report the average $95\%$ mean confidence interval in brackets/errorbars. In the imbalanced support set experiments, we evaluate tasks with various imbalance levels and distributions, as specified in figures and tables. In the imbalanced meta-dataset experiments, we evaluate using regular, balanced 5-shot 5-way tasks.

**Data Augmentation.**    During the meta-/pre-training phases, we apply standard data augmentation techniques, following a similar setup to Chen et al. (2019), with a random rotation of 10 degrees, scaling, random color/contrast/brightness jitter. Meta-validation and meta-testing had no augmentation apart from in the *Random-Shot (ROS+)* setting where the same augmentations were applied on the oversampled support images. All images are resized to 84 by 84 pixels.

## A.3   BACKBONE ARCHITECTURES

All methods shared the same backbone architecture. For the core contribution of our work, we used Conv4 architecture consisting of 4 convolutional layers with 64 channels (padding 1), interleaved by batch normalization (Ioffe & Szegedy, 2015), ReLU activation function, and max-pooling (kernel size 2, and stride 2) (Chen et al., 2019). Relation Network used max-pooling only for the last 2 layers of the backbone to account for the Relation Module. The Relation Module consisted of two additional convolutional layers, each followed by batch norm, ReLU, max-pooling).

For experiments with different backbones: Conv6, ResNet10, and ResNet34 (Chen et al., 2019). Conv6 extended the Conv4 backbone to 6 convolutional layers, and max-pooling applied after each first 4 layers. ResNet models (He et al., 2016) followed the same setup as Chen et al. (2019).

For imbalanced meta-dataset and imbalanced reduced meta-dataset, we used the Conv4 model, with 32 channels instead of 64, due to less training data.

## A.4   FSL METHODS AND BASELINES

In our experiments, we used a wide range of FSL methods (full details can be found in our source code):

1. **Baseline (fine-tune)** (Pan & Yang, 2010) represents a classical way of applying transfer learning, where a neural network is pre-trained on a large dataset, then fine-tuned on a smaller domain-specific dataset. The baseline's backbone followed a single linear classification layer with a single output for each meta-training dataset class. The whole network was trained during pre-training. During meta-testing, the baseline's pre-trained linear layer was exchanged for another randomly initialized classification layer with outputs matching the task's number of classes ($N$-way). Fine-tuning was performed on the new randomly initialized classification layer using the support set $\mathcal{S}$.

2. **Baseline (1-NN)** is another classical method of applying transfer learning but using a k-nearest neighbor classifier instead of the classification layer during meta-validation. Pre-training was performed in the same was the fine-tune baselines. During the meta-testing time, instead of fine-tuning, the model matched query samples to the nearest support sample's class based on Euclidian distance.

3. **Baseline++** (Chen et al., 2019) augments the fine-tune baseline by using Cosine Similarity on the last layer.

4. **Matching Network (Matching Net)** (Vinyals et al., 2017) uses context embeddings with an LSTM to effectively perform k-nearest neighbor in embedding space using cosine similarity to classify the query set.

5. **Prototypical Networks (ProtoNet)** (Snell et al., 2017) maps images into a feature space and calculates class means (called prototypes). The query samples are then classified based on the closest Euclidian distance to a classes' prototype. We evaluate two models, one meta-trained like the others on 5-way episodes, and another variation trained on 20-way episodes. During 20-way meta-training, we set the class' query size to 5.

6. **Relation Networks (Relation Net)** (Sung et al., 2017) augment the classical Prototypical Networks by introducing a relation module (another neural network) that compares the distance instead of using Euclidian. The original method uses Mean Squared Error to minimize the relation score between samples of the same type. However, we follow work by Chen et al. (2019), use cross-entropy loss that expedites meta-training. The structure of the relation module is described in section A.3.

7. **DKT** (formally called GPShot) proposed by Patacchiola et al. (2020) is a probabilistic approach that utilizes the Gaussian Processes with a deep neural network as a kernel function. We used Batch Norm Cosine distance for the kernel type.

8. **SimpleShot** (Wang et al., 2019a) augments the 1-NN baseline model by normalizing and centering the feature vector using the dataset's mean feature vector. The query samples are assigned to the nearest prototype's class according to Euclidian distance. In contrast to the baseline models, pre-training is performed on the meta-training dataset like other meta-learning algorithms, and meta-validation is used to select the best model based on performance on tasks sampled from $\mathcal{D}_{val}$.

9. **MAML** (Finn et al., 2017) is a meta-learning technique that learns a common initialization of weights that can be quickly adapted for task using fine-tuning on the support set. The task adaptation process uses a standard gradient descent algorithm minimizing Cross-Entropy loss on the support set. The original method uses second-order derivates; however, due to more efficient calculation, we use the first-order MAML, which has been shown to work just as well. We set the inner-learning rate to 0.1 with 10 iteration steps. We optimize the meta-learner model on batches of 4 meta-training tasks. These hyperparameters were selected based on our hyperparameter fine-tuning.

10. **ProtoMAML** (Triantafillou et al., 2020) augments traditional first-order MAML by reinitializing the last classification layer between tasks. Specifically, the weights of the layer are assigned to the prototype for each class's corresponding output. This extra step combines the fine-tuning ability of MAML and the class regularisation ability of Prototypical Networks. We set the inner-loop learning rate to 0.1 with 10 iterations. Unlike for MAML, we found that updating the meta-learner after a single meta-training task gave the best performance.

11. **Bayesian-MAML (BMAML)** (Yoon et al., 2018) augments the MAML method by replacing the inner loop's standard stochastic gradient descent with Bayesian gradient-based updates. BMAML uses Stein Variational Gradient Descent that is a non-parametric variational inference method combining strengths of Monte Carlo approximation and variational inference. The algorithm learns to approximate a posterior over the initialization parameters conditioned on the task support set. Yoon et al. (2018) also adds a **chaser** loss, which utilizes the samples in the query set during meta-training to approximate the true task posterior. Minimization of the KL divergence between the true task posterior and the estimated task parameter posterior can be used to drive the meta-training process. We set the inner-loop learning rate to 0.1 with 1 inner-loop step. We found that using a higher inner-loop step number destabilized performance. We used 20 particles. The chaser loss variation used a learning rate of 0.5. Again, we found these combinations of hyperparameters to give the best results.

12. **Bayesian-TAML (BTAML)** (Lee et al., 2019) *[Left out of the main paper body.]* This method augments the MAML method with four main changes: 1) task-dependent parameter initialization $z$, 2) task-dependent per-layer learning rate multipliers $\gamma$, 3) task-dependent per-class learning rate multiplier $\omega$, 4) meta-learned per-parameter learning rate $\alpha$ (similar to Meta-SGD, Li et al. (2017)). However, our results for BTAML were unstable, suggesting a fault in our implementation. Unfortunately, we did not identify it in time for the submission, and we decided to move the method's results to the appendix. In our experiments, we explored several variations to this method with various parameters $(z, \omega, \gamma, \alpha)$ turned on and off, as well as different hyperparameters. We found that using a meta-learning rate of $10^{-4}$ performed better than $10^{-3}$ in contrast to the other models. We set the inner-learning rate to 0.1 with 10 iteration steps, and optimize the meta-learner model on batches of 4 meta-training tasks. Again, we found this set up worked best in our experiments.

## A.5 CLASS IMBALANCE TECHNIQUES AND STRATEGIES

We pair FSL methods with popular data-level class-imbalance strategies:

1. **Random Over-Sampling (ROS)** (Japkowicz & Stephen, 2002) without and with data augmentation (**ROS** and **ROS+**). For the augmentations we used: random sized cropping between 0.15 and 1.1 scale of the image, random horizontal flip, and random color/brightness/contrast jitter. A visualization of ROS and ROS+ is presented in Figure 7. During meta-training, ROS+ augmentations were applied twice: once when sampling from the meta-training dataset, and the second time during the support set resampling. This may have slightly destabilized meta-training, which would explain why sometimes Random-

Shot with ROS (ROS+ at inference only) achieved better performance than Random-Shot with ROS+ in Figures 3 and 4.

2. **Random-Shot Meta-Training** (Triantafillou et al., 2020; Lee et al., 2019; Chen et al., 2020) was applied as specified in the main body of the paper (Section 3.4).

3. **Focal Loss** (Lin et al., 2017). Focal Loss has been found to be very effective in combating the class-imbalance problem on the one-stage object detectors. We exchanged the inner-loop cross-entropy loss of optimization-based algorithms and fine-tune baselines with the focal loss with $\gamma = 2$ and $\alpha = 1$. Results are presented in Figure 6 in the main paper body and in Appendix D.1.

4. **Weighted Loss**. Weighted loss is also commonly used to rebalance the effects of class-imbalance (Buda et al., 2018; Leevy et al., 2018). We weight the inner-loop cross-entropy loss of optimization-based algorithms and fine-tune baselines by inverse class frequency of support set samples. Results are presented in Figure 6 in the main paper body and in Appendix D.1.

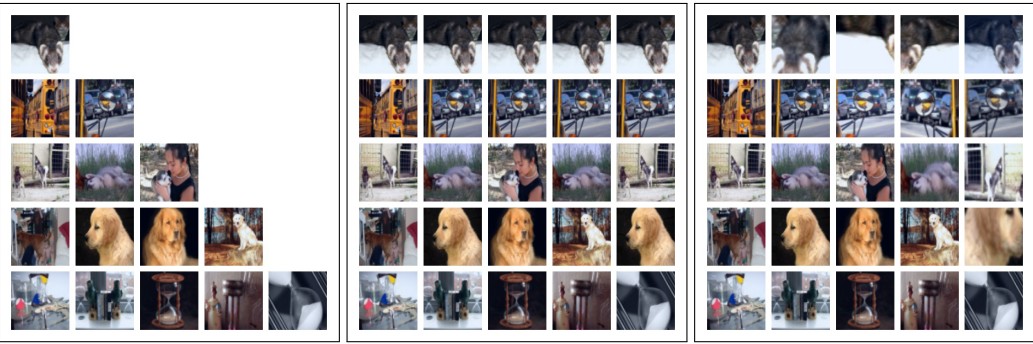

Figure 7: Visualisation of linear 1-5shot support sets. **Left:** no ROS. **Middle:** ROS. **Right:** ROS+.

# B  VERIFICATION OF IMPLEMENTATION

We implement the FSL methods in PyTorch, adapting the implementation of (Chen et al., 2019) but also borrowing from other implementations online (see individual method files in the source code for individual attribution). However, we have heavily modified these implementations to fit our imbalanced FSL framework, which also offers standard and continual FSL compatibility (Antoniou et al., 2020). We provide our implementations for ProtoMAML and BTAML for which no open-source implementation in PyTorch existed as of writing. To verify our implementations, we compare methods on the standard balanced 5-shot 5-way task with reported accuracy. Results are presented in Table 2. We see that algorithms achieve very similar performance with no less than 3% accuracy points compared to the reported performance. The discrepancies can be accounted for due to smaller training batch for SimpleShot, different augmentation strategies for the other methods, and natural variance stemming from random initialization. We show the validation performance over epochs for each method in Figure 8 on the next page.

Table 2: Results of standard 5-shot 5-way experiments on Mini-ImageNet as achieved with our implementation compared to the original (reported) accuracy and other work. Other Sources's Accuracies were taken from: * (Chen et al., 2019), † (Snell & Zemel, 2020), ‡ (Vogelbaum et al., 2020)

| Model | Our Acc (95%CI) | Original Acc(95%CI) | Other Sources' Acc(95%CI) |
|---|---|---|---|
| Baseline (1-NN) | $39.72_{\pm0.73}$ | - | - |
| Baseline (fine-tune) | $62.67_{\pm0.70}$ | $62.53_{\pm0.69}$ | - |
| Baseline++ | $66.43_{\pm0.66}$ | $66.43_{\pm0.63}$ | - |
| Matching Net | $62.27_{\pm0.69}$ | $55.31_{\pm0.73}$ | $63.48_{\pm0.66}$ * |
| ProtoNet | $64.37_{\pm0.71}$ | $65.77_{\pm0.70}$ | $64.24_{\pm0.72}$ * |
| ProtoNet (20-way) | $65.76_{\pm0.70}$ | $68.20_{\pm0.66}$ | $66.68_{\pm0.68}$ * |
| Relation Net (CE) | $64.76_{\pm0.68}$ | $65.32_{\pm0.70}$ | $66.60_{\pm0.69}$ * |
| DKT | $62.92_{\pm0.67}$ | $64.00_{\pm0.09}$ | $62.88_{\pm0.46}$ † |
| SimpleShot | $63.74_{\pm0.69}$ | $66.92_{\pm0.17}$ | - |
| MAML | $61.83_{\pm0.71}$ | $63.15_{\pm0.91}$ | $62.71_{\pm0.71}$ * |
| ProtoMAML | $59.86_{\pm0.76}$ | - | $60.70_{\pm0.99}$ ‡ |
| BMAML | $59.89_{\pm0.68}$ | - | $59.23_{\pm0.34}$ † |
| BMAML (chaser) | $56.45_{\pm0.67}$ | - | $59.93_{\pm0.31}$ † |

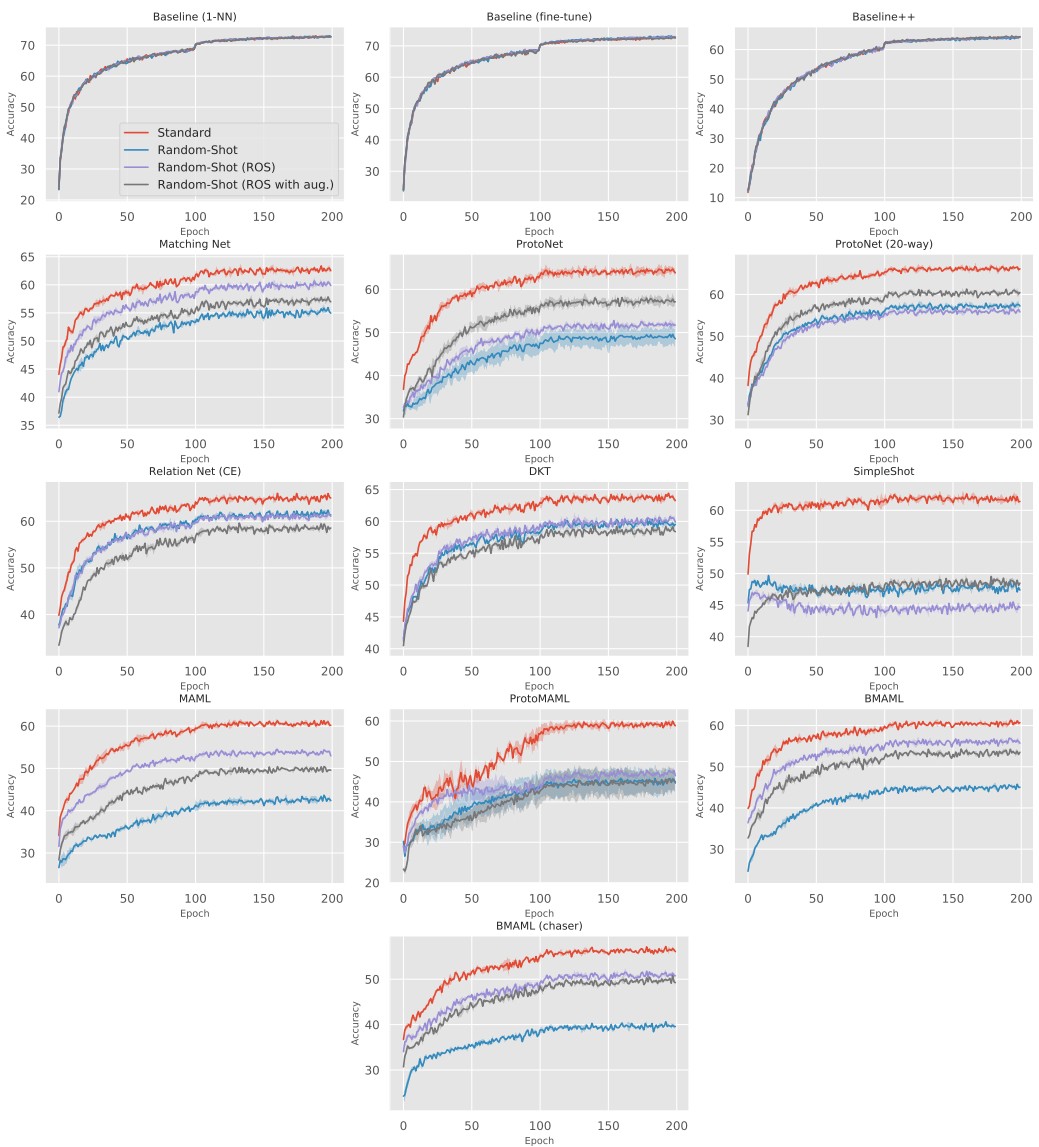

Figure 8: Validation performance through epochs on Standard 5-shot 5-way meta-training, and Random Shot (1-9shot random). The shaded areas show $\pm$ 1 standard deviation over three repeats on different seeds.

# C BREAKDOWN OF RESULTS

In this section, we breakdown the results from the main body of the paper. Specifically, we provide the top-50 performing models on the imbalanced 1-9shot linear task in subsection C.1. The breakdown of higher shots experiment (Figure 4) is provided in subsection C.2. In subsection C.3, we include the breakdown of the backbone experiment (Figure 5) showing the performance by algorithm type and training procedure. We provide precision and recall tables of linear 1-9shot 5-way tasks for each meta-training procedure in subsection C.4. In section C.5, we provide the results for the imbalanced reduced meta-dataset (Figure 10).

## C.1 TOP-50 PERFORMING MODELS ON 1-9SHOT LINEAR

Table 3: Top-50 models using different meta-training strategies, showing absolute and relative difference between the balanced and the imbalanced task. Results sorted by relative difference.

| Model | Training Method | 5shot | 1-9shot linear | Abs. Diff. | Rel. Diff |
|---|---|---|---|---|---|
| Matching Net | Random-Shot (ROS+ at inference) | $58.12_{\pm0.67}$ | $56.17_{\pm0.69}$ | -1.94861 | -0.0335301 |
|  | Random-Shot with ROS (ROS+ at inference) | $60.59_{\pm0.68}$ | $58.30_{\pm0.69}$ | -2.28958 | -0.0377858 |
|  | Standard (ROS+ at inference) | $60.26_{\pm0.66}$ | $57.90_{\pm0.67}$ | -2.36208 | -0.0391952 |
| Relation Net (CE) | Random-Shot with ROS+ | $60.48_{\pm0.71}$ | $57.99_{\pm0.72}$ | -2.49653 | -0.0412758 |
| ProtoNet (20-way) | Random-Shot with ROS+ | $61.21_{\pm0.72}$ | $58.58_{\pm0.69}$ | -2.62778 | -0.0429288 |
| Matching Net | Random-Shot with ROS | $61.75_{\pm0.70}$ | $59.07_{\pm0.72}$ | -2.67431 | -0.0433091 |
| Relation Net (CE) | Random-Shot | $63.50_{\pm0.70}$ | $60.64_{\pm0.71}$ | -2.86181 | -0.0450648 |
| Matching Net | Random-Shot with ROS+ | $60.05_{\pm0.68}$ | $57.24_{\pm0.69}$ | -2.81111 | -0.0468096 |
| Baseline (1-NN) | Standard | $39.72_{\pm0.73}$ | $37.85_{\pm0.72}$ | -1.87222 | -0.0471337 |
| BMAML | Random-Shot (ROS+ at inference) | $49.97_{\pm0.67}$ | $47.53_{\pm0.66}$ | -2.43611 | -0.0487513 |
| Baseline (1-NN) | Random-Shot with ROS | $39.90_{\pm0.75}$ | $37.95_{\pm0.73}$ | -1.95139 | -0.0489112 |
| BMAML (chaser) | Random-Shot (ROS+ at inference) | $45.00_{\pm0.62}$ | $42.79_{\pm0.63}$ | -2.20833 | -0.0490748 |
| Relation Net (CE) | Random-Shot with ROS (ROS+ at inference) | $62.97_{\pm0.70}$ | $59.84_{\pm0.71}$ | -3.13403 | -0.0497668 |
|  | Random-Shot with ROS | $64.12_{\pm0.71}$ | $60.93_{\pm0.71}$ | -3.19444 | -0.0498181 |
| DKT | Random-Shot (ROS+ at inference) | $62.33_{\pm0.66}$ | $59.17_{\pm0.67}$ | -3.15486 | -0.0506178 |
| Relation Net (CE) | Random-Shot (ROS+ at inference) | $62.53_{\pm0.71}$ | $59.30_{\pm0.71}$ | -3.22361 | -0.0515549 |
| Baseline++ | Random-Shot (ROS+ at inference) | $65.52_{\pm0.66}$ | $62.14_{\pm0.68}$ | -3.37847 | -0.0515628 |
| DKT | Random-Shot with ROS (ROS+ at inference) | $62.40_{\pm0.67}$ | $59.18_{\pm0.68}$ | -3.22083 | -0.0516159 |
| Baseline++ | Random-Shot with ROS (ROS+ at inference) | $65.00_{\pm0.66}$ | $61.60_{\pm0.69}$ | -3.40694 | -0.0524117 |
| Baseline (1-NN) | Random-Shot | $40.83_{\pm0.74}$ | $38.68_{\pm0.72}$ | -2.14236 | -0.0524749 |
| ProtoNet | Random-Shot with ROS+ | $60.05_{\pm0.71}$ | $56.89_{\pm0.68}$ | -3.15764 | -0.0525823 |
| BMAML | Standard (ROS+ at inference) | $58.82_{\pm0.68}$ | $55.72_{\pm0.70}$ | -3.10458 | -0.0527811 |
|  | Random-Shot with ROS (ROS+ at inference) | $58.00_{\pm0.68}$ | $54.84_{\pm0.69}$ | -3.15486 | -0.0543948 |
| Baseline++ | Standard (ROS+ at inference) | $65.27_{\pm0.66}$ | $61.72_{\pm0.68}$ | -3.55583 | -0.0544764 |
| DKT | Standard (ROS+ at inference) | $61.96_{\pm0.66}$ | $58.57_{\pm0.67}$ | -3.3925 | -0.054749 |
|  | Random-Shot with ROS+ | $61.16_{\pm0.67}$ | $57.71_{\pm0.70}$ | -3.44722 | -0.0563623 |
| MAML | Random-Shot (ROS+ at inference) | $50.39_{\pm0.69}$ | $47.52_{\pm0.68}$ | -2.86597 | -0.0568763 |
| BMAML (chaser) | Random-Shot with ROS | $53.52_{\pm0.64}$ | $50.46_{\pm0.68}$ | -3.05417 | -0.057068 |
| ProtoNet | Random-Shot (ROS+ at inference) | $50.67_{\pm0.68}$ | $47.76_{\pm0.66}$ | -2.90972 | -0.0574303 |
| BMAML | Random-Shot with ROS+ | $56.52_{\pm0.69}$ | $53.23_{\pm0.71}$ | -3.28403 | -0.0581087 |
| ProtoNet | Random-Shot | $51.65_{\pm0.68}$ | $48.57_{\pm0.65}$ | -3.08194 | -0.0596698 |
| BMAML (chaser) | Random-Shot with ROS (ROS+ at inference) | $51.87_{\pm0.62}$ | $48.74_{\pm0.65}$ | -3.13194 | -0.0603821 |
| ProtoNet (20-way) | Random-Shot (ROS+ at inference) | $58.31_{\pm0.72}$ | $54.79_{\pm0.69}$ | -3.52153 | -0.0603935 |
| Baseline (1-NN) | Random-Shot with ROS (ROS+ at inference) | $39.22_{\pm0.69}$ | $36.83_{\pm0.68}$ | -2.39028 | -0.0609506 |
| MAML | Random-Shot with ROS+ | $54.60_{\pm0.72}$ | $51.25_{\pm0.74}$ | -3.35069 | -0.0613626 |
|  | Random-Shot with ROS (ROS+ at inference) | $58.36_{\pm0.72}$ | $54.70_{\pm0.72}$ | -3.66319 | -0.0627707 |
| Relation Net (CE) | Standard (ROS+ at inference) | $63.89_{\pm0.69}$ | $59.85_{\pm0.69}$ | -4.035 | -0.0631558 |
| BMAML (chaser) | Standard (ROS+ at inference) | $55.40_{\pm0.65}$ | $51.89_{\pm0.68}$ | -3.50958 | -0.0633513 |
| ProtoNet (20-way) | Random-Shot with ROS (ROS+ at inference) | $58.32_{\pm0.70}$ | $54.62_{\pm0.70}$ | -3.70556 | -0.0635359 |
| BMAML | Random-Shot with ROS | $58.98_{\pm0.68}$ | $55.23_{\pm0.72}$ | -3.75139 | -0.0636038 |
| Baseline (1-NN) | Standard (ROS+ at inference) | $39.75_{\pm0.71}$ | $37.18_{\pm0.68}$ | -2.57042 | -0.0646625 |
| MAML | Standard (ROS+ at inference) | $61.00_{\pm0.71}$ | $57.04_{\pm0.72}$ | -3.96083 | -0.0649286 |
| ProtoNet | Random-Shot with ROS (ROS+ at inference) | $54.39_{\pm0.69}$ | $50.74_{\pm0.66}$ | -3.64375 | -0.0669961 |
| Baseline (fine-tune) | Random-Shot with ROS+ | $60.46_{\pm0.70}$ | $56.12_{\pm0.69}$ | -4.33819 | -0.0717551 |
| DKT | Random-Shot with ROS | $63.21_{\pm0.67}$ | $58.65_{\pm0.68}$ | -4.55625 | -0.0720799 |
| Baseline (1-NN) | Random-Shot (ROS+ at inference) | $39.98_{\pm0.71}$ | $37.08_{\pm0.70}$ | -2.90417 | -0.0726382 |
| Baseline (fine-tune) | Random-Shot with ROS (ROS+ at inference) | $61.82_{\pm0.69}$ | $57.03_{\pm0.67}$ | -4.79236 | -0.0775149 |
|  | Standard (ROS+ at inference) | $61.46_{\pm0.71}$ | $56.65_{\pm0.68}$ | -4.8075 | -0.0782216 |
| ProtoMAML | Random-Shot (ROS+ at inference) | $54.09_{\pm0.71}$ | $49.73_{\pm0.71}$ | -4.35903 | -0.0805921 |
| Baseline (fine-tune) | Random-Shot (ROS+ at inference) | $61.63_{\pm0.70}$ | $56.55_{\pm0.66}$ | -5.07569 | -0.0823633 |

## C.2 RANDOM 1-29SHOT AND 1-49SHOT TASKS

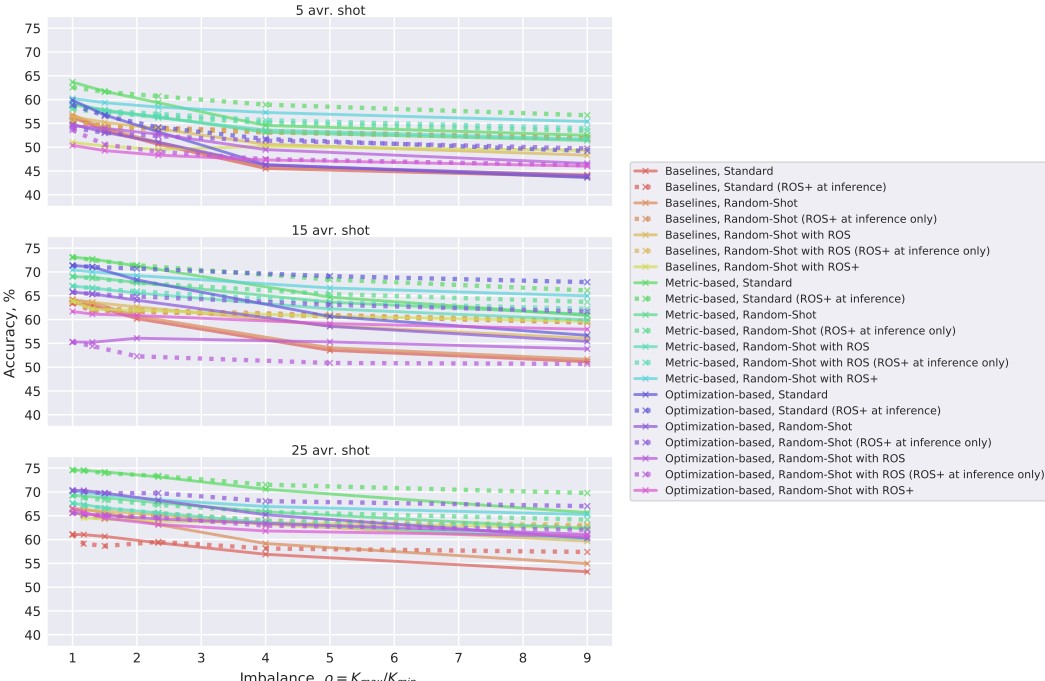

Figure 9: Linear imbalance by model type. Best viewed in color on a computer. The metric-based methods (represented by green-cyan lines) tend to be perform better than the baselines (red-orange) and the optimization-based models (purple-pink).

## C.3 BACKBONE EXPERIMENTS

We run additional experiments with different backbone models. In Tables 4 and 5, we show the random 1-9 shot 5-way performance on Conv6, ResNet10, and ResNet34. We can observe that for many of the methods, ROS still benefits the models. However, the reader should exercise caution since we used the same hyper-parameterization as for the four-layered CNN. We did not perform any hyperparameter fine-tuning on these backbones; the results would likely be higher if we allowed for longer meta-training. Some results are missing due to destabilization in meta-training caused by deeper backbones.

Table 4: Standard vs Random Shot (accuracy)

| | Standard | | | Random-Shot | | |
|---|---|---|---|---|---|---|
| | Conv6 | ResNet10 | ResNet34 | Conv6 | ResNet10 | ResNet34 |
| Baseline (1-NN) | $51.21_{\pm0.74}$ | $53.18_{\pm0.78}$ | $53.25_{\pm0.75}$ | $50.01_{\pm0.70}$ | $52.24_{\pm0.74}$ | $52.49_{\pm0.77}$ |
| Baseline (fine-tune) | $54.12_{\pm0.79}$ | $58.55_{\pm0.87}$ | $59.02_{\pm0.86}$ | $54.62_{\pm0.80}$ | $58.36_{\pm0.86}$ | $60.54_{\pm0.85}$ |
| Baseline++ | $51.95_{\pm0.83}$ | $50.94_{\pm0.84}$ | $54.18_{\pm0.86}$ | $52.23_{\pm0.83}$ | $50.99_{\pm0.83}$ | $52.91_{\pm0.80}$ |
| Matching Net | $54.99_{\pm0.70}$ | $58.60_{\pm0.72}$ | $60.00_{\pm0.74}$ | $53.70_{\pm0.73}$ | - | $56.34_{\pm0.73}$ |
| ProtoNet | $\mathbf{57.92}_{\pm0.78}$ | $\mathbf{62.06}_{\pm0.74}$ | $\mathbf{64.35}_{\pm0.74}$ | $56.31_{\pm0.72}$ | $60.00_{\pm0.73}$ | $59.81_{\pm0.73}$ |
| ProtoNet (20-way) | $57.66_{\pm0.76}$ | $60.61_{\pm0.80}$ | - | - | $\mathbf{62.17}_{\pm0.79}$ | $\mathbf{62.46}_{\pm0.77}$ |
| DKT | $55.60_{\pm0.70}$ | $57.98_{\pm0.73}$ | $57.07_{\pm0.75}$ | $\mathbf{57.44}_{\pm0.72}$ | $58.77_{\pm0.70}$ | - |
| SimpleShot | $54.36_{\pm0.88}$ | $60.95_{\pm0.81}$ | $60.48_{\pm0.88}$ | $54.90_{\pm0.81}$ | $61.33_{\pm0.80}$ | $55.25_{\pm0.74}$ |
| MAML | $52.53_{\pm0.77}$ | $57.69_{\pm0.79}$ | $52.52_{\pm0.74}$ | $46.91_{\pm0.72}$ | $52.61_{\pm0.77}$ | - |
| ProtoMAML | $53.00_{\pm0.75}$ | $53.59_{\pm0.86}$ | $51.22_{\pm0.85}$ | $49.72_{\pm0.76}$ | $54.37_{\pm0.74}$ | $52.48_{\pm0.72}$ |
| BMAML | $38.54_{\pm0.83}$ | $34.61_{\pm0.84}$ | $39.39_{\pm0.85}$ | $47.82_{\pm0.76}$ | $48.09_{\pm0.72}$ | $46.76_{\pm0.71}$ |
| BMAML (chaser) | $35.81_{\pm0.71}$ | $30.90_{\pm0.57}$ | $30.27_{\pm0.57}$ | $33.03_{\pm0.54}$ | $31.62_{\pm0.59}$ | $26.41_{\pm0.48}$ |

Table 5: Random Shot (ROS) vs Random Shot (ROS+) (accuracy)

| | Random-Shot (ROS) | | | Random-Shot (ROS with aug.) | | |
|---|---|---|---|---|---|---|
| | Conv6 | ResNet10 | ResNet34 | Conv6 | ResNet10 | ResNet34 |
| Baseline (1-NN) | $51.05_{\pm0.75}$ | $52.65_{\pm0.75}$ | - | $46.75_{\pm0.79}$ | $50.69_{\pm0.76}$ | - |
| Baseline (fine-tune) | $55.32_{\pm0.80}$ | $59.77_{\pm0.88}$ | $61.51_{\pm0.83}$ | $54.48_{\pm0.77}$ | $59.02_{\pm0.80}$ | $60.48_{\pm0.80}$ |
| Baseline++ | $\mathbf{61.08}_{\pm0.75}$ | $57.41_{\pm0.74}$ | $58.07_{\pm0.75}$ | $56.95_{\pm0.78}$ | $54.81_{\pm0.77}$ | $55.49_{\pm0.81}$ |
| Matching Net | $58.32_{\pm0.72}$ | $\mathbf{60.48}_{\pm0.72}$ | - | $56.24_{\pm0.74}$ | $59.89_{\pm0.70}$ | $55.19_{\pm0.71}$ |
| ProtoNet | $56.99_{\pm0.79}$ | $60.47_{\pm0.73}$ | - | $58.60_{\pm0.72}$ | $\mathbf{62.05}_{\pm0.76}$ | - |
| ProtoNet (20-way) | $57.66_{\pm0.77}$ | $59.85_{\pm0.73}$ | $\mathbf{61.60}_{\pm0.73}$ | $\mathbf{58.82}_{\pm0.72}$ | - | $\mathbf{61.78}_{\pm0.78}$ |
| DKT | $57.01_{\pm0.72}$ | $59.12_{\pm0.71}$ | - | $56.36_{\pm0.73}$ | $58.09_{\pm0.73}$ | - |
| SimpleShot | $52.99_{\pm0.82}$ | $60.46_{\pm0.80}$ | - | $51.44_{\pm0.79}$ | $54.20_{\pm0.83}$ | $37.14_{\pm0.69}$ |
| MAML | $56.25_{\pm0.74}$ | $59.72_{\pm0.76}$ | - | $50.27_{\pm0.78}$ | $46.76_{\pm0.74}$ | - |
| ProtoMAML | $56.99_{\pm0.77}$ | $55.96_{\pm0.79}$ | $45.96_{\pm0.70}$ | $41.45_{\pm0.69}$ | $47.17_{\pm0.75}$ | $40.85_{\pm0.77}$ |
| BMAML | $56.58_{\pm0.74}$ | $58.20_{\pm0.74}$ | $61.02_{\pm0.76}$ | $53.55_{\pm0.70}$ | $57.25_{\pm0.75}$ | $60.50_{\pm0.74}$ |
| BMAML (chaser) | $52.23_{\pm0.69}$ | $35.18_{\pm0.64}$ | $32.33_{\pm0.60}$ | $47.76_{\pm0.71}$ | $38.84_{\pm0.66}$ | $23.68_{\pm0.42}$ |

## C.4 PRECISION AND RECALL TABLES

Table 6: Precision and recall for linear 1-9shot 5-way tasks after **Standard** meta-training.

| | Precision(95%CI) | | | | | Recall(95%CI) | | | | | Avr. F1(95%CI) |
|---|---|---|---|---|---|---|---|---|---|---|---|
| | $K_1=1$ | $K_2=3$ | $K_3=5$ | $K_4=7$ | $K_5=9$ | $K_1=1$ | $K_2=3$ | $K_3=5$ | $K_4=7$ | $K_5=9$ | - |
| Baseline (1-NN) | $0.32_{\pm0.32}$ | $0.46_{\pm0.25}$ | $0.50_{\pm0.19}$ | $0.48_{\pm0.15}$ | $0.45_{\pm0.13}$ | $0.13_{\pm0.10}$ | $0.29_{\pm0.19}$ | $0.41_{\pm0.20}$ | $0.50_{\pm0.21}$ | $0.57_{\pm0.20}$ | $0.32_{\pm0.02}$ |
| Baseline (fine-tune) | $0.05_{\pm0.10}$ | $0.63_{\pm0.34}$ | $0.63_{\pm0.08}$ | $0.50_{\pm0.04}$ | $0.41_{\pm0.03}$ | $0.00_{\pm0.00}$ | $0.19_{\pm0.08}$ | $0.53_{\pm0.12}$ | $0.75_{\pm0.07}$ | $0.90_{\pm0.02}$ | $0.39_{\pm0.01}$ |
| Baseline++ | $0.02_{\pm0.04}$ | $0.45_{\pm0.44}$ | $\mathbf{0.66_{\pm0.13}}$ | $0.52_{\pm0.05}$ | $0.39_{\pm0.02}$ | $0.00_{\pm0.00}$ | $0.13_{\pm0.08}$ | $0.48_{\pm0.17}$ | $0.76_{\pm0.09}$ | $0.94_{\pm0.01}$ | $0.37_{\pm0.01}$ |
| Matching Net | $0.54_{\pm0.39}$ | $0.67_{\pm0.13}$ | $0.62_{\pm0.06}$ | $0.55_{\pm0.04}$ | $0.48_{\pm0.03}$ | $0.15_{\pm0.08}$ | $0.41_{\pm0.14}$ | $0.61_{\pm0.12}$ | $0.73_{\pm0.08}$ | $0.84_{\pm0.04}$ | $0.50_{\pm0.02}$ |
| ProtoNet | $0.18_{\pm0.28}$ | $0.70_{\pm0.15}$ | $0.62_{\pm0.07}$ | $0.54_{\pm0.06}$ | $0.51_{\pm0.05}$ | $0.03_{\pm0.01}$ | $0.37_{\pm0.12}$ | $0.65_{\pm0.09}$ | $0.78_{\pm0.06}$ | $0.84_{\pm0.03}$ | $0.47_{\pm0.01}$ |
| ProtoNet (20-way) | $0.10_{\pm0.18}$ | $\mathbf{0.71_{\pm0.21}}$ | $0.64_{\pm0.08}$ | $0.55_{\pm0.06}$ | $0.49_{\pm0.05}$ | $0.01_{\pm0.00}$ | $0.31_{\pm0.12}$ | $0.63_{\pm0.10}$ | $\mathbf{0.79_{\pm0.06}}$ | $0.88_{\pm0.03}$ | $0.45_{\pm0.01}$ |
| Relation Net (CE) | $0.42_{\pm0.43}$ | $0.67_{\pm0.10}$ | $0.62_{\pm0.06}$ | $0.56_{\pm0.05}$ | $0.54_{\pm0.05}$ | $0.09_{\pm0.04}$ | $\mathbf{0.48_{\pm0.12}}$ | $\mathbf{0.68_{\pm0.09}}$ | $0.76_{\pm0.06}$ | $0.81_{\pm0.04}$ | $0.51_{\pm0.01}$ |
| DKT | $\mathbf{0.57_{\pm0.26}}$ | $0.63_{\pm0.09}$ | $0.61_{\pm0.06}$ | $\mathbf{0.57_{\pm0.05}}$ | $\mathbf{0.55_{\pm0.04}}$ | $\mathbf{0.22_{\pm0.11}}$ | $0.48_{\pm0.13}$ | $0.65_{\pm0.11}$ | $0.73_{\pm0.08}$ | $0.79_{\pm0.06}$ | $\mathbf{0.53_{\pm0.02}}$ |
| SimpleShot | $0.01_{\pm0.02}$ | $0.59_{\pm0.41}$ | $0.66_{\pm0.09}$ | $0.51_{\pm0.06}$ | $0.41_{\pm0.04}$ | $0.00_{\pm0.00}$ | $0.15_{\pm0.07}$ | $0.52_{\pm0.12}$ | $0.77_{\pm0.06}$ | $0.90_{\pm0.02}$ | $0.38_{\pm0.01}$ |
| MAML | $0.00_{\pm0.01}$ | $0.65_{\pm0.19}$ | $0.58_{\pm0.07}$ | $0.48_{\pm0.03}$ | $0.42_{\pm0.02}$ | $0.00_{\pm0.00}$ | $0.26_{\pm0.08}$ | $0.55_{\pm0.10}$ | $0.75_{\pm0.06}$ | $0.85_{\pm0.03}$ | $0.41_{\pm0.01}$ |
| ProtoMAML | $0.18_{\pm0.26}$ | $0.59_{\pm0.28}$ | $0.55_{\pm0.16}$ | $0.44_{\pm0.11}$ | $0.38_{\pm0.08}$ | $0.03_{\pm0.02}$ | $0.23_{\pm0.10}$ | $0.49_{\pm0.16}$ | $0.68_{\pm0.20}$ | $0.76_{\pm0.21}$ | $0.36_{\pm0.02}$ |
| BMAML | $0.00_{\pm0.00}$ | $0.06_{\pm0.10}$ | $0.38_{\pm0.38}$ | $0.51_{\pm0.14}$ | $0.27_{\pm0.01}$ | $0.00_{\pm0.00}$ | $0.01_{\pm0.00}$ | $0.16_{\pm0.12}$ | $0.51_{\pm0.22}$ | $0.98_{\pm0.00}$ | $0.22_{\pm0.01}$ |
| BMAML (chaser) | $0.00_{\pm0.00}$ | $0.00_{\pm0.00}$ | $0.15_{\pm0.22}$ | $0.46_{\pm0.20}$ | $0.24_{\pm0.00}$ | $0.00_{\pm0.00}$ | $0.00_{\pm0.00}$ | $0.04_{\pm0.03}$ | $0.37_{\pm0.20}$ | $\mathbf{0.99_{\pm0.00}}$ | $0.16_{\pm0.01}$ |

Table 7: Precision and recall for linear 1-9shot 5-way tasks after **Random Shot** meta-training.

| | Precision(95%CI) | | | | | Recall(95%CI) | | | | | Avr. F1(95%CI) |
|---|---|---|---|---|---|---|---|---|---|---|---|
| | $K_1=1$ | $K_2=3$ | $K_3=5$ | $K_4=7$ | $K_5=9$ | $K_1=1$ | $K_2=3$ | $K_3=5$ | $K_4=7$ | $K_5=9$ | - |
| Baseline (1-NN) | $0.32_{\pm0.32}$ | $0.48_{\pm0.26}$ | $0.51_{\pm0.19}$ | $0.49_{\pm0.15}$ | $0.46_{\pm0.13}$ | $0.13_{\pm0.10}$ | $0.30_{\pm0.18}$ | $0.42_{\pm0.20}$ | $0.50_{\pm0.20}$ | $0.58_{\pm0.19}$ | $0.33_{\pm0.02}$ |
| Baseline (fine-tune) | $0.06_{\pm0.11}$ | $0.63_{\pm0.33}$ | $0.63_{\pm0.08}$ | $0.50_{\pm0.04}$ | $0.41_{\pm0.02}$ | $0.00_{\pm0.00}$ | $0.19_{\pm0.08}$ | $0.53_{\pm0.12}$ | $0.76_{\pm0.06}$ | $0.90_{\pm0.02}$ | $0.39_{\pm0.01}$ |
| Baseline++ | $0.02_{\pm0.03}$ | $0.45_{\pm0.44}$ | $\mathbf{0.67_{\pm0.13}}$ | $0.52_{\pm0.05}$ | $0.39_{\pm0.02}$ | $0.00_{\pm0.00}$ | $0.13_{\pm0.08}$ | $0.49_{\pm0.17}$ | $\mathbf{0.77_{\pm0.08}}$ | $\mathbf{0.94_{\pm0.01}}$ | $0.37_{\pm0.01}$ |
| Matching Net | $\mathbf{0.58_{\pm0.27}}$ | $\mathbf{0.65_{\pm0.17}}$ | $0.62_{\pm0.08}$ | $0.55_{\pm0.05}$ | $0.45_{\pm0.03}$ | $0.24_{\pm0.12}$ | $0.36_{\pm0.15}$ | $0.55_{\pm0.15}$ | $0.68_{\pm0.11}$ | $0.84_{\pm0.04}$ | $0.49_{\pm0.02}$ |
| ProtoNet | $0.47_{\pm0.28}$ | $0.51_{\pm0.09}$ | $0.51_{\pm0.06}$ | $0.49_{\pm0.05}$ | $0.48_{\pm0.05}$ | $0.17_{\pm0.08}$ | $0.44_{\pm0.11}$ | $0.57_{\pm0.09}$ | $0.61_{\pm0.08}$ | $0.63_{\pm0.07}$ | $0.45_{\pm0.01}$ |
| ProtoNet (20-way) | $0.52_{\pm0.35}$ | $0.60_{\pm0.09}$ | $0.58_{\pm0.06}$ | $0.55_{\pm0.06}$ | $0.54_{\pm0.05}$ | $0.16_{\pm0.07}$ | $0.50_{\pm0.11}$ | $\mathbf{0.65_{\pm0.08}}$ | $0.70_{\pm0.07}$ | $0.73_{\pm0.06}$ | $0.51_{\pm0.01}$ |
| Relation Net (CE) | $0.55_{\pm0.09}$ | $0.62_{\pm0.07}$ | $0.64_{\pm0.06}$ | $\mathbf{0.63_{\pm0.06}}$ | $\mathbf{0.64_{\pm0.06}}$ | $\mathbf{0.48_{\pm0.13}}$ | $\mathbf{0.58_{\pm0.11}}$ | $0.64_{\pm0.09}$ | $0.66_{\pm0.09}$ | $0.67_{\pm0.08}$ | $\mathbf{0.59_{\pm0.01}}$ |
| DKT | $0.58_{\pm0.27}$ | $0.63_{\pm0.09}$ | $0.61_{\pm0.06}$ | $0.57_{\pm0.04}$ | $0.55_{\pm0.04}$ | $0.21_{\pm0.11}$ | $0.48_{\pm0.14}$ | $0.65_{\pm0.11}$ | $0.74_{\pm0.08}$ | $0.79_{\pm0.05}$ | $0.53_{\pm0.01}$ |
| SimpleShot | $0.06_{\pm0.10}$ | $0.64_{\pm0.30}$ | $0.62_{\pm0.08}$ | $0.50_{\pm0.05}$ | $0.44_{\pm0.04}$ | $0.01_{\pm0.00}$ | $0.22_{\pm0.09}$ | $0.56_{\pm0.11}$ | $0.76_{\pm0.06}$ | $0.87_{\pm0.03}$ | $0.41_{\pm0.01}$ |
| MAML | $0.07_{\pm0.13}$ | $0.45_{\pm0.30}$ | $0.50_{\pm0.08}$ | $0.42_{\pm0.03}$ | $0.36_{\pm0.01}$ | $0.01_{\pm0.00}$ | $0.16_{\pm0.07}$ | $0.44_{\pm0.13}$ | $0.64_{\pm0.10}$ | $0.81_{\pm0.04}$ | $0.33_{\pm0.01}$ |
| ProtoMAML | $0.15_{\pm0.24}$ | $0.59_{\pm0.22}$ | $0.55_{\pm0.10}$ | $0.45_{\pm0.06}$ | $0.39_{\pm0.04}$ | $0.02_{\pm0.01}$ | $0.23_{\pm0.07}$ | $0.49_{\pm0.11}$ | $0.68_{\pm0.10}$ | $0.79_{\pm0.09}$ | $0.37_{\pm0.01}$ |
| BMAML | $0.14_{\pm0.22}$ | $0.42_{\pm0.32}$ | $0.51_{\pm0.15}$ | $0.44_{\pm0.06}$ | $0.37_{\pm0.03}$ | $0.03_{\pm0.02}$ | $0.19_{\pm0.13}$ | $0.44_{\pm0.21}$ | $0.62_{\pm0.18}$ | $0.82_{\pm0.08}$ | $0.33_{\pm0.02}$ |
| BMAML (chaser) | $0.06_{\pm0.10}$ | $0.31_{\pm0.29}$ | $0.44_{\pm0.15}$ | $0.40_{\pm0.05}$ | $0.35_{\pm0.02}$ | $0.01_{\pm0.00}$ | $0.13_{\pm0.09}$ | $0.36_{\pm0.19}$ | $0.60_{\pm0.16}$ | $0.78_{\pm0.09}$ | $0.28_{\pm0.01}$ |

Table 8: Precision and recall for linear 1-9shot 5-way after **Random Shot (ROS)** meta-training.

| | Precision(95%CI) | | | | | Recall(95%CI) | | | | | Avr. F1(95%CI) |
|---|---|---|---|---|---|---|---|---|---|---|---|
| | $K_1=1$ | $K_2=3$ | $K_3=5$ | $K_4=7$ | $K_5=9$ | $K_1=1$ | $K_2=3$ | $K_3=5$ | $K_4=7$ | $K_5=9$ | - |
| Baseline (1-NN) | $0.31_{\pm0.31}$ | $0.47_{\pm0.25}$ | $0.50_{\pm0.20}$ | $0.49_{\pm0.15}$ | $0.46_{\pm0.13}$ | $0.13_{\pm0.10}$ | $0.29_{\pm0.18}$ | $0.42_{\pm0.21}$ | $0.50_{\pm0.20}$ | $0.57_{\pm0.20}$ | $0.32_{\pm0.02}$ |
| Baseline (fine-tune) | $0.16_{\pm0.26}$ | $0.69_{\pm0.22}$ | $0.62_{\pm0.08}$ | $0.52_{\pm0.04}$ | $0.44_{\pm0.03}$ | $0.02_{\pm0.01}$ | $0.26_{\pm0.10}$ | $0.57_{\pm0.11}$ | $0.76_{\pm0.06}$ | $\mathbf{0.88_{\pm0.02}}$ | $0.42_{\pm0.01}$ |
| Baseline++ | $0.49_{\pm0.45}$ | $\mathbf{0.71_{\pm0.11}}$ | $0.64_{\pm0.06}$ | $0.57_{\pm0.04}$ | $0.52_{\pm0.04}$ | $0.12_{\pm0.06}$ | $0.45_{\pm0.14}$ | $0.66_{\pm0.11}$ | $0.78_{\pm0.07}$ | $0.86_{\pm0.03}$ | $0.52_{\pm0.02}$ |
| Matching Net | $0.56_{\pm0.11}$ | $0.60_{\pm0.07}$ | $0.62_{\pm0.06}$ | $0.61_{\pm0.05}$ | $0.61_{\pm0.05}$ | $0.46_{\pm0.14}$ | $0.57_{\pm0.11}$ | $0.62_{\pm0.11}$ | $0.64_{\pm0.10}$ | $0.66_{\pm0.09}$ | $0.57_{\pm0.02}$ |
| ProtoNet | $0.49_{\pm0.32}$ | $0.55_{\pm0.10}$ | $0.54_{\pm0.07}$ | $0.51_{\pm0.05}$ | $0.49_{\pm0.04}$ | $0.15_{\pm0.07}$ | $0.44_{\pm0.11}$ | $0.58_{\pm0.09}$ | $0.65_{\pm0.07}$ | $0.71_{\pm0.06}$ | $0.47_{\pm0.01}$ |
| ProtoNet (20-way) | $0.51_{\pm0.36}$ | $0.60_{\pm0.11}$ | $0.58_{\pm0.07}$ | $0.54_{\pm0.06}$ | $0.52_{\pm0.05}$ | $0.15_{\pm0.07}$ | $0.47_{\pm0.12}$ | $0.62_{\pm0.09}$ | $0.69_{\pm0.07}$ | $0.75_{\pm0.05}$ | $0.50_{\pm0.02}$ |
| Relation Net (CE) | $0.56_{\pm0.09}$ | $0.62_{\pm0.07}$ | $0.64_{\pm0.06}$ | $\mathbf{0.63_{\pm0.06}}$ | $\mathbf{0.64_{\pm0.05}}$ | $0.48_{\pm0.13}$ | $\mathbf{0.58_{\pm0.11}}$ | $0.64_{\pm0.09}$ | $0.66_{\pm0.09}$ | $0.69_{\pm0.07}$ | $\mathbf{0.60_{\pm0.01}}$ |
| DKT | $\mathbf{0.59_{\pm0.21}}$ | $0.63_{\pm0.08}$ | $0.61_{\pm0.05}$ | $0.59_{\pm0.05}$ | $0.57_{\pm0.04}$ | $0.27_{\pm0.12}$ | $0.51_{\pm0.13}$ | $0.66_{\pm0.10}$ | $0.72_{\pm0.08}$ | $0.77_{\pm0.06}$ | $0.55_{\pm0.02}$ |
| SimpleShot | $0.10_{\pm0.17}$ | $0.61_{\pm0.27}$ | $0.60_{\pm0.09}$ | $0.50_{\pm0.05}$ | $0.41_{\pm0.03}$ | $0.01_{\pm0.00}$ | $0.22_{\pm0.09}$ | $0.50_{\pm0.11}$ | $0.71_{\pm0.07}$ | $0.86_{\pm0.03}$ | $0.39_{\pm0.01}$ |
| MAML | $0.52_{\pm0.34}$ | $0.59_{\pm0.10}$ | $0.56_{\pm0.05}$ | $0.52_{\pm0.04}$ | $0.49_{\pm0.03}$ | $0.15_{\pm0.06}$ | $0.43_{\pm0.11}$ | $0.60_{\pm0.09}$ | $0.70_{\pm0.06}$ | $0.76_{\pm0.05}$ | $0.49_{\pm0.01}$ |
| ProtoMAML | $0.35_{\pm0.39}$ | $0.58_{\pm0.16}$ | $0.53_{\pm0.12}$ | $0.46_{\pm0.12}$ | $0.44_{\pm0.10}$ | $0.07_{\pm0.03}$ | $0.37_{\pm0.12}$ | $0.54_{\pm0.15}$ | $0.63_{\pm0.20}$ | $0.72_{\pm0.17}$ | $0.40_{\pm0.02}$ |
| BMAML | $0.49_{\pm0.09}$ | $0.57_{\pm0.10}$ | $0.59_{\pm0.09}$ | $0.61_{\pm0.10}$ | $0.63_{\pm0.10}$ | $\mathbf{0.54_{\pm0.17}}$ | $0.55_{\pm0.17}$ | $0.58_{\pm0.17}$ | $0.55_{\pm0.18}$ | $0.54_{\pm0.18}$ | $0.52_{\pm0.02}$ |
| BMAML (chaser) | $0.45_{\pm0.08}$ | $0.52_{\pm0.09}$ | $0.54_{\pm0.09}$ | $0.55_{\pm0.10}$ | $0.57_{\pm0.12}$ | $0.51_{\pm0.17}$ | $0.51_{\pm0.17}$ | $0.54_{\pm0.17}$ | $0.50_{\pm0.18}$ | $0.47_{\pm0.18}$ | $0.47_{\pm0.02}$ |

Table 9: Precision and recall for linear 1-9shot 5-way after **Random Shot (ROS+)** meta-training.

| | Precision(95%CI) | | | | | Recall(95%CI) | | | | | Avr. F1(95%CI) |
|---|---|---|---|---|---|---|---|---|---|---|---|
| | $K_1=1$ | $K_2=3$ | $K_3=5$ | $K_4=7$ | $K_5=9$ | $K_1=1$ | $K_2=3$ | $K_3=5$ | $K_4=7$ | $K_5=9$ | - |
| Baseline (1-NN) | $0.36_{\pm0.09}$ | $0.33_{\pm0.05}$ | $0.42_{\pm0.39}$ | $0.46_{\pm0.38}$ | $0.48_{\pm0.35}$ | $\mathbf{0.57_{\pm0.19}}$ | $\mathbf{0.73_{\pm0.13}}$ | $0.10_{\pm0.05}$ | $0.13_{\pm0.07}$ | $0.15_{\pm0.08}$ | $0.26_{\pm0.01}$ |
| Baseline (fine-tune) | $0.60_{\pm0.19}$ | $0.56_{\pm0.08}$ | $0.63_{\pm0.07}$ | $0.58_{\pm0.05}$ | $0.55_{\pm0.05}$ | $0.28_{\pm0.10}$ | $0.52_{\pm0.11}$ | $0.53_{\pm0.10}$ | $0.69_{\pm0.07}$ | $0.78_{\pm0.05}$ | $0.53_{\pm0.01}$ |
| Baseline++ | $0.61_{\pm0.15}$ | $\mathbf{0.61_{\pm0.11}}$ | $0.64_{\pm0.08}$ | $0.60_{\pm0.05}$ | $0.54_{\pm0.04}$ | $0.37_{\pm0.14}$ | $0.47_{\pm0.15}$ | $0.54_{\pm0.14}$ | $0.72_{\pm0.09}$ | $\mathbf{0.82_{\pm0.05}}$ | $0.55_{\pm0.02}$ |
| Matching Net | $0.55_{\pm0.10}$ | $0.60_{\pm0.09}$ | $0.62_{\pm0.06}$ | $0.59_{\pm0.05}$ | $0.56_{\pm0.04}$ | $0.43_{\pm0.14}$ | $0.47_{\pm0.13}$ | $0.58_{\pm0.12}$ | $0.66_{\pm0.10}$ | $0.72_{\pm0.08}$ | $0.55_{\pm0.02}$ |
| ProtoNet | $0.59_{\pm0.19}$ | $0.55_{\pm0.07}$ | $0.56_{\pm0.05}$ | $0.59_{\pm0.06}$ | $0.63_{\pm0.06}$ | $0.28_{\pm0.10}$ | $0.58_{\pm0.11}$ | $0.67_{\pm0.08}$ | $0.68_{\pm0.07}$ | $0.64_{\pm0.07}$ | $0.55_{\pm0.01}$ |
| ProtoNet (20-way) | $\mathbf{0.62_{\pm0.23}}$ | $0.56_{\pm0.07}$ | $0.57_{\pm0.06}$ | $\mathbf{0.62_{\pm0.06}}$ | $\mathbf{0.68_{\pm0.07}}$ | $0.25_{\pm0.10}$ | $0.63_{\pm0.10}$ | $\mathbf{0.72_{\pm0.07}}$ | $0.69_{\pm0.07}$ | $0.63_{\pm0.08}$ | $\mathbf{0.56_{\pm0.01}}$ |
| Relation Net (CE) | $0.55_{\pm0.12}$ | $0.57_{\pm0.07}$ | $0.60_{\pm0.06}$ | $0.60_{\pm0.06}$ | $0.62_{\pm0.06}$ | $0.41_{\pm0.14}$ | $0.54_{\pm0.12}$ | $0.63_{\pm0.10}$ | $0.66_{\pm0.09}$ | $0.66_{\pm0.08}$ | $0.56_{\pm0.02}$ |
| DKT | $0.56_{\pm0.13}$ | $0.61_{\pm0.10}$ | $0.62_{\pm0.06}$ | $0.58_{\pm0.05}$ | $0.57_{\pm0.04}$ | $0.37_{\pm0.13}$ | $0.45_{\pm0.13}$ | $0.61_{\pm0.12}$ | $0.70_{\pm0.09}$ | $0.76_{\pm0.06}$ | $0.55_{\pm0.02}$ |
| SimpleShot | $0.50_{\pm0.38}$ | $0.56_{\pm0.12}$ | $0.58_{\pm0.09}$ | $0.52_{\pm0.06}$ | $0.52_{\pm0.06}$ | $0.11_{\pm0.04}$ | $0.41_{\pm0.11}$ | $0.51_{\pm0.12}$ | $\mathbf{0.76_{\pm0.06}}$ | $0.79_{\pm0.04}$ | $0.47_{\pm0.01}$ |
| MAML | $0.48_{\pm0.10}$ | $0.47_{\pm0.07}$ | $0.52_{\pm0.07}$ | $0.53_{\pm0.05}$ | $0.56_{\pm0.05}$ | $0.38_{\pm0.10}$ | $0.47_{\pm0.09}$ | $0.54_{\pm0.09}$ | $0.59_{\pm0.08}$ | $0.60_{\pm0.07}$ | $0.50_{\pm0.01}$ |
| ProtoMAML | $0.43_{\pm0.11}$ | $0.43_{\pm0.10}$ | $0.47_{\pm0.10}$ | $0.48_{\pm0.09}$ | $0.51_{\pm0.09}$ | $0.32_{\pm0.09}$ | $0.44_{\pm0.13}$ | $0.48_{\pm0.14}$ | $0.53_{\pm0.14}$ | $0.53_{\pm0.14}$ | $0.43_{\pm0.02}$ |
| BMAML | $0.48_{\pm0.09}$ | $0.57_{\pm0.16}$ | $0.60_{\pm0.14}$ | $0.59_{\pm0.10}$ | $0.56_{\pm0.07}$ | $0.54_{\pm0.18}$ | $0.41_{\pm0.18}$ | $0.47_{\pm0.19}$ | $0.57_{\pm0.19}$ | $0.68_{\pm0.15}$ | $0.49_{\pm0.02}$ |
| BMAML (chaser) | $0.44_{\pm0.08}$ | $0.51_{\pm0.11}$ | $0.57_{\pm0.11}$ | $0.56_{\pm0.08}$ | $0.54_{\pm0.06}$ | $0.48_{\pm0.16}$ | $0.41_{\pm0.15}$ | $0.46_{\pm0.17}$ | $0.55_{\pm0.16}$ | $0.63_{\pm0.14}$ | $0.47_{\pm0.02}$ |

## C.5 Imbalanced Reduced Meta-Training Dataset

In this section, we present the results for the reduced meta-training MiniImageNet dataset (with 32 classes).

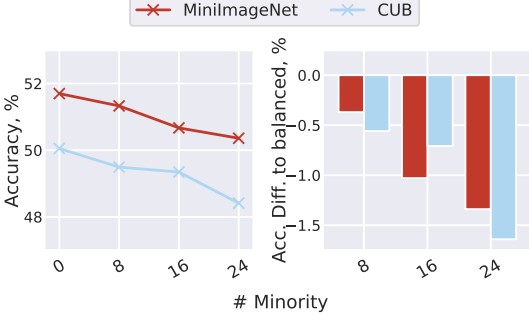

Figure 10: Combined model average performance with increasing minority classes. *Left:* Combined accuracy of all models and training scenarios. *Right:* Relative performance difference to the balanced dataset.

Table 10: Table showing full results for the reduced meta-training MiniImageNet dataset (with 32 classes), evaluated on (meta-test split of) MiniImageNet. The last two columns show the average and maximum model's (absolute) performance difference to the balanced task. RelationNet suffers the most from the imbalanced meta-training.

| Imbalance $\mathcal{I}$ | balanced | | step | | Avr. Diff. | Max. Diff. |
|---|---|---|---|---|---|---|
| Max. # class samples | 150 | 190 | 270 | 510 | - | - |
| Min. # class samples | 150 | 30 | 30 | 30 | - | - |
| # Minority | - | 16 | 32 | 48 | - | - |
| Baseline (1-NN) | $40.46_{\pm0.65}$ | $41.86_{\pm0.66}$ | $40.93_{\pm0.67}$ | $40.52_{\pm0.66}$ | 0.5 | 0.0 |
| Baseline (fine-tune) | $54.13_{\pm0.70}$ | $53.72_{\pm0.68}$ | $53.36_{\pm0.69}$ | $52.45_{\pm0.72}$ | -0.7 | -1.7 |
| Baseline++ | $54.40_{\pm0.64}$ | $54.15_{\pm0.65}$ | $53.55_{\pm0.64}$ | $52.89_{\pm0.65}$ | -0.7 | -1.5 |
| Matching Net | $53.56_{\pm0.68}$ | $53.34_{\pm0.69}$ | $52.69_{\pm0.67}$ | $51.62_{\pm0.67}$ | -0.8 | -1.9 |
| ProtoNet | $54.14_{\pm0.69}$ | $53.54_{\pm0.70}$ | $53.38_{\pm0.69}$ | $52.81_{\pm0.70}$ | -0.7 | -1.3 |
| ProtoNet (20-way) | $55.05_{\pm0.69}$ | $54.98_{\pm0.70}$ | $53.40_{\pm0.70}$ | $51.98_{\pm0.69}$ | -1.2 | -3.1 |
| Relation Net (CE) | $53.83_{\pm0.68}$ | $52.80_{\pm0.69}$ | $49.97_{\pm0.66}$ | $49.55_{\pm0.67}$ | -2.3 | -4.3 |
| DKT | $54.09_{\pm0.66}$ | $53.57_{\pm0.64}$ | $53.01_{\pm0.66}$ | $52.27_{\pm0.67}$ | -0.9 | -1.8 |
| SimpleShot | $56.05_{\pm0.71}$ | $55.96_{\pm0.71}$ | $55.70_{\pm0.71}$ | $54.83_{\pm0.71}$ | -0.4 | -1.2 |
| MAML | $50.40_{\pm0.73}$ | $49.95_{\pm0.71}$ | $49.27_{\pm0.72}$ | $49.07_{\pm0.72}$ | -0.7 | -1.3 |
| ProtoMAML | $42.57_{\pm0.66}$ | $40.75_{\pm0.68}$ | $42.10_{\pm0.67}$ | $45.96_{\pm0.68}$ | 0.3 | -1.8 |

Table 11: Table showing full results for the reduced meta-training MiniImageNet dataset (with 32 classes), evaluated on (meta-test split of) CUB. The last two columns show the average and maximum model's (absolute) performance difference to the balanced task. RelationNet suffers the most from the imbalanced meta-training dataset.

| Imbalance $\mathcal{I}$ | balanced | | step | | Avr. Diff. | Max. Diff. |
|---|---|---|---|---|---|---|
| Max. # class samples | 150 | 190 | 270 | 510 | - | - |
| Min. # class samples | 150 | 30 | 30 | 30 | - | - |
| # Minority | - | 16 | 32 | 48 | - | - |
| Baseline (1-NN) | $41.87_{\pm0.66}$ | $41.85_{\pm0.66}$ | $41.38_{\pm0.66}$ | $40.42_{\pm0.67}$ | -0.5 | -1.4 |
| Baseline (fine-tune) | $53.16_{\pm0.69}$ | $52.63_{\pm0.69}$ | $52.98_{\pm0.69}$ | $50.69_{\pm0.69}$ | -0.8 | -2.5 |
| Baseline++ | $51.65_{\pm0.69}$ | $52.40_{\pm0.69}$ | $52.31_{\pm0.72}$ | $48.67_{\pm0.67}$ | -0.4 | -3.0 |
| Matching Net | $49.89_{\pm0.72}$ | $49.12_{\pm0.70}$ | $49.12_{\pm0.69}$ | $47.82_{\pm0.69}$ | -0.9 | -2.1 |
| ProtoNet | $50.01_{\pm0.71}$ | $49.78_{\pm0.72}$ | $49.34_{\pm0.71}$ | $48.97_{\pm0.71}$ | -0.5 | -1.0 |
| ProtoNet (20-way) | $50.09_{\pm0.72}$ | $49.40_{\pm0.72}$ | $49.16_{\pm0.71}$ | $47.74_{\pm0.67}$ | -1.0 | -2.4 |
| Relation Net (CE) | $50.71_{\pm0.71}$ | $50.31_{\pm0.69}$ | $48.68_{\pm0.69}$ | $46.78_{\pm0.66}$ | -1.6 | -3.9 |
| DKT | $53.84_{\pm0.71}$ | $52.45_{\pm0.69}$ | $53.75_{\pm0.70}$ | $52.42_{\pm0.69}$ | -0.7 | -1.4 |
| SimpleShot | $53.01_{\pm0.72}$ | $52.60_{\pm0.70}$ | $52.03_{\pm0.71}$ | $50.44_{\pm0.70}$ | -1.0 | -2.6 |
| MAML | $51.15_{\pm0.71}$ | $50.67_{\pm0.70}$ | $50.23_{\pm0.71}$ | $49.90_{\pm0.69}$ | -0.7 | -1.2 |
| ProtoMAML | $45.20_{\pm0.68}$ | $43.22_{\pm0.69}$ | $43.81_{\pm0.68}$ | $48.69_{\pm0.68}$ | 0.0 | -2.0 |

# D ADDITIONAL EXPERIMENTS

## D.1 ADDITIONAL IMBALANCE STRATEGIES

In Figure 11, we compare Standard meta-training and Random-Shot meta-training with focal loss. We observe no significant advantage of using Focal Loss Random-Shot meta-training over the Standard meta-training experiments.

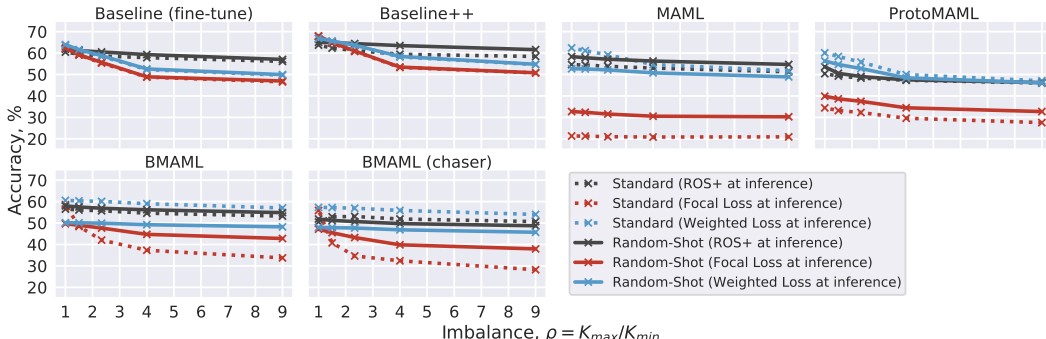

Figure 11: Standard episodic (meta-)training (Vinyals et al., 2017) and random-shot episodic meta-training (Triantafillou et al., 2020) with Weighted Loss (Buda et al., 2018) and Focal Loss (Lin et al., 2017), applied to the inner-loop of optimization-based functions and fine-tune baselines.

## D.2 BTAML

We implemented and trained Bayesian TAML (Lee et al., 2019); however, the training performance graphs suggest a mistake in our implementation, which we did not manage to identify in time for the submission. For this reason, we have left out these results from the main paper. The BTAML $(\alpha, \omega, \gamma, z)$ corresponds to full BTAML, while others indicate variants with the corresponding components turn off. We provide models' performances with the full performance in Appendix E.

Table 12: Performance of our implementation of BTAML on 5shot 5-way task. The BTAML $(\alpha, \omega, \gamma, z)$ indicates the full version of the proposed model.

| Model | Acc (95%CI) | F1 (95%CI) |
|---|---|---|
| BTAML $(\alpha, \omega, \gamma, z)$ | $52.94_{\pm 0.76}$ | $52.30_{\pm 1.43}$ |
| BTAML $(\alpha, \gamma, z)$ | $\mathbf{54.89}_{\pm 0.75}$ | $\mathbf{54.52}_{\pm 1.39}$ |
| BTAML $(\alpha)$ | $52.19_{\pm 0.76}$ | $51.85_{\pm 1.41}$ |

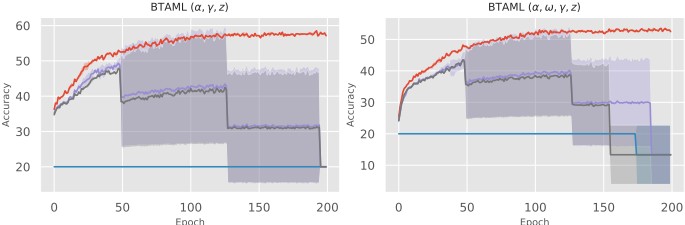

Figure 12: Validation performance of BTAML through epochs using Standard 5-shot 5-way meta-training, and Random Shot (1-9shot random). The shaded areas show $\pm$ 1 standard deviation over three repeats on different seeds.

## D.3 ANALYSIS OF SAMPLES PER CLASS IN META-TRAINING DATASET

Table 13: Meta-/Pre- Training with reduced number of samples in the meta-training dataset of Mini-ImageNet (all 64 classes are balanced). Setting with 600 * samples uses 64 channels for each 4 convolutional layers instead of 32 channels as was done for the other '#Class Samples' settings. In addition, all three Baselines were trained using conventional split on $D_{train}$ instead of $D'_{train}$. The table suggests that the number of samples per class in the meta-training dataset is not very significant on the performance of FSL algorithms beyond a certain point. All settings were trained on 50k tasks, apart from 600 * trained on 100k tasks.

| # Class Samples Model | 50 | 100 | 300 | 600 * |
|---|---|---|---|---|
| Baseline (1-NN) | $38.75_{\pm 0.61}$ | $39.82_{\pm 0.60}$ | $42.69_{\pm 0.66}$ | $39.72_{\pm 0.73}$ |
| Baseline (fine-tune) | $44.34_{\pm 0.68}$ | $46.10_{\pm 0.69}$ | $51.26_{\pm 0.70}$ | $62.67_{\pm 0.70}$ |
| Baseline++ | $41.63_{\pm 0.64}$ | $49.51_{\pm 0.68}$ | $48.44_{\pm 0.65}$ | $\mathbf{66.43}_{\pm \mathbf{0.66}}$ |
| Matching Net | $55.90_{\pm 0.70}$ | $58.30_{\pm 0.70}$ | $58.26_{\pm 0.68}$ | $62.27_{\pm 0.69}$ |
| ProtoNet | $\mathbf{58.43}_{\pm \mathbf{0.69}}$ | $60.09_{\pm 0.70}$ | $60.65_{\pm 0.70}$ | $64.37_{\pm 0.71}$ |
| ProtoNet (20-way) | $57.87_{\pm 0.72}$ | $59.88_{\pm 0.70}$ | $60.91_{\pm 0.70}$ | $65.76_{\pm 0.70}$ |
| Relation Net (CE) | $56.50_{\pm 0.70}$ | $\mathbf{60.93}_{\pm \mathbf{0.68}}$ | $\mathbf{62.78}_{\pm \mathbf{0.70}}$ | $64.76_{\pm 0.68}$ |
| DKT | $56.27_{\pm 0.65}$ | $57.93_{\pm 0.69}$ | $58.09_{\pm 0.69}$ | $62.92_{\pm 0.67}$ |

# E    FULL RESULTS FOR MAIN EXPERIMENTS

Figures 14 and 13 show accuracy and F1 scores, respectively, on imbalanced tasks of Standard (5-shot) and Random Shot (1-9shot) meta-training. Their corresponding result tables are in Tables 14 to 21. Table 22 provides the full results for the main imbalanced meta-dataset experiments.

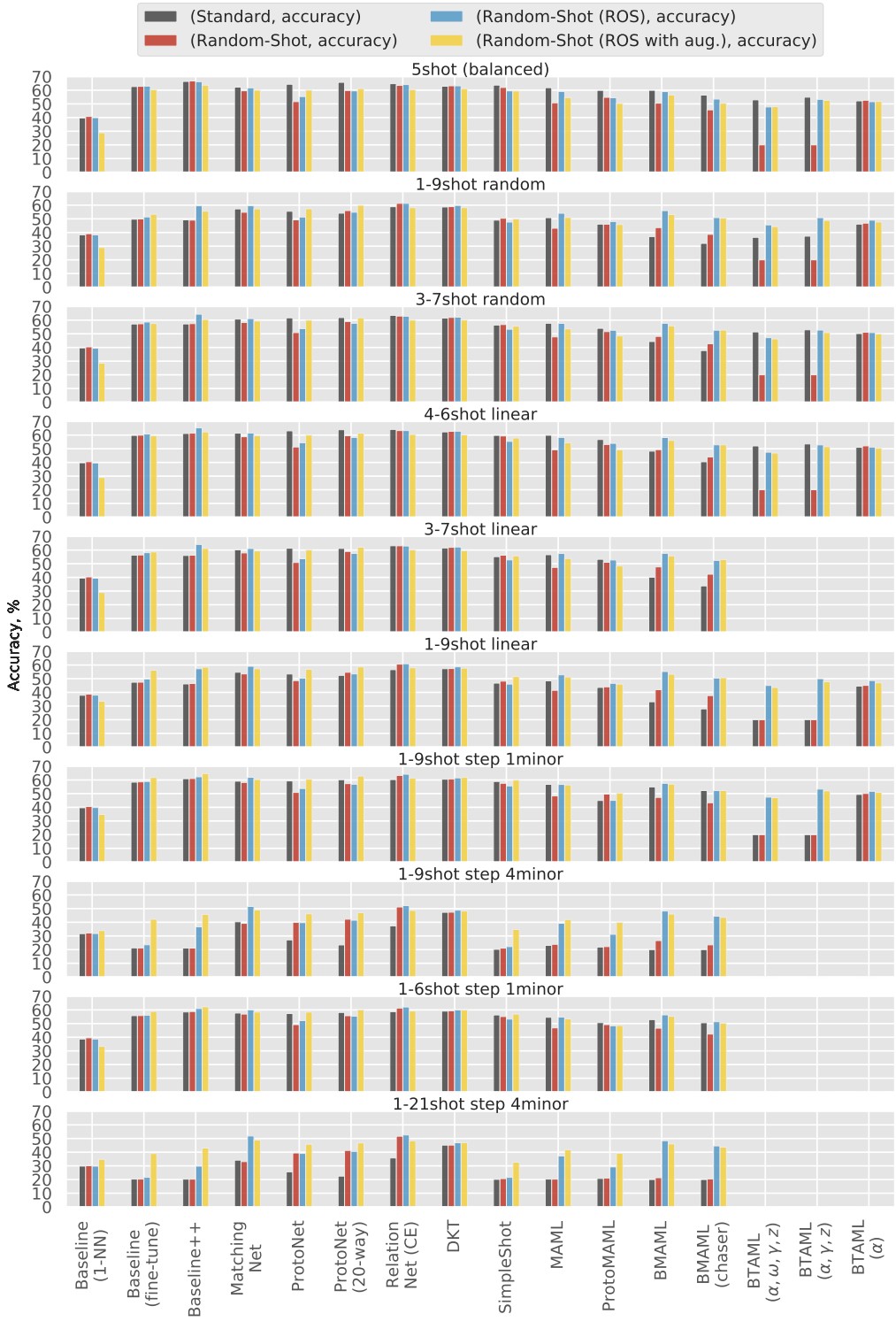

Figure 13: Accuracy

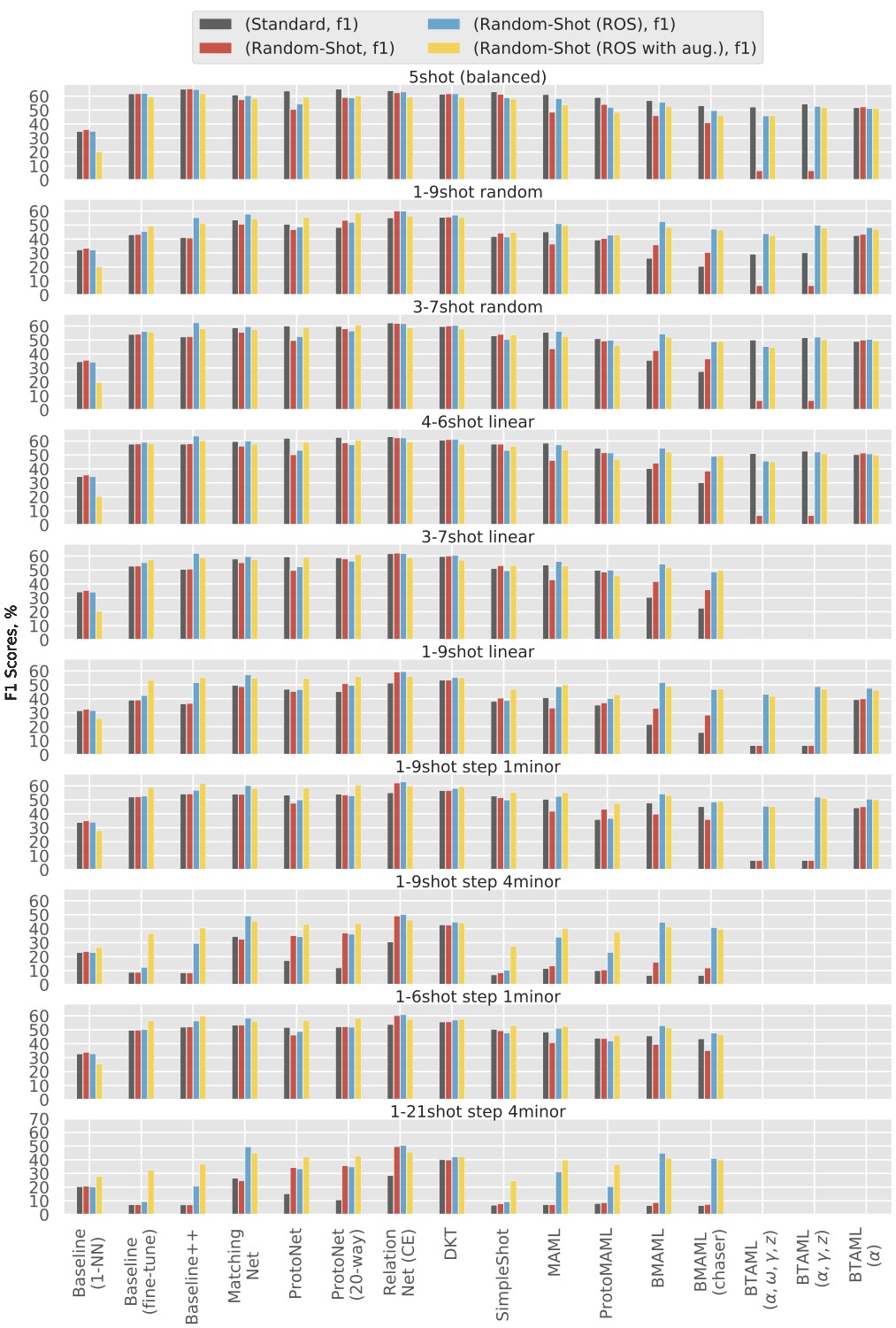

Figure 14: F1 Scores.

Table 14: Standard (accuracy)

| | 5shot balanced | 1-9shot random | 3-7shot random | 4-6shot linear | 1-9shot linear | 1-9shot step 1minor | 1-9shot step 4minor | 1-21shot step 4minor | Rank |
|---|---|---|---|---|---|---|---|---|---|
| Baseline (1-NN) | 39.72±0.73 | 38.30±0.71 | 39.68±0.73 | 39.72±0.73 | 37.85±0.72 | 39.67±0.69 | 31.68±0.68 | 29.81±0.65 | 11.6 |
| Baseline (fine-tune) | 62.67±0.70 | 49.72±0.85 | 57.17±0.74 | 59.78±0.70 | 47.40±0.59 | 58.39±0.57 | 21.26±0.18 | 20.33±0.08 | 7.8 |
| Baseline++ | **66.43±0.66** | 49.28±0.89 | 57.20±0.76 | 61.17±0.69 | 46.16±0.59 | **60.93±0.54** | 21.17±0.21 | 20.28±0.09 | 6.5 |
| Matching Net | 62.27±0.69 | 57.26±0.73 | 60.77±0.69 | 61.41±0.68 | 54.69±0.65 | 59.09±0.60 | 40.35±0.71 | 34.17±0.64 | 4.4 |
| ProtoNet | 64.37±0.71 | 55.54±0.82 | 61.68±0.73 | 63.10±0.70 | 53.42±0.60 | 59.28±0.57 | 27.11±0.50 | 25.64±0.44 | 4.1 |
| ProtoNet (20-way) | 65.76±0.70 | 54.14±0.87 | 61.86±0.72 | 63.92±0.71 | 52.43±0.61 | 60.18±0.56 | 23.42±0.35 | 22.50±0.29 | 4.0 |
| Relation Net (CE) | 64.76±0.68 | **58.86±0.78** | **63.50±0.70** | **64.14±0.69** | 56.59±0.63 | 60.28±0.58 | 37.36±0.70 | 35.77±0.67 | **2.0** |
| DKT | 62.92±0.67 | 58.66±0.73 | 61.52±0.69 | 62.32±0.67 | **57.30±0.67** | 60.54±0.63 | **47.27±0.71** | **45.18±0.71** | 2.6 |
| SimpleShot | 63.74±0.69 | 49.05±0.89 | 56.41±0.77 | 59.77±0.70 | 46.75±0.60 | 58.89±0.56 | 20.22±0.06 | 20.13±0.04 | 8.6 |
| MAML | 61.83±0.71 | 50.83±0.80 | 57.70±0.70 | 59.87±0.69 | 48.41±0.60 | 56.77±0.59 | 23.09±0.30 | 20.32±0.07 | 7.4 |
| ProtoMAML | 59.86±0.76 | 46.14±0.92 | 54.00±0.87 | 56.76±0.81 | 43.62±0.86 | 45.00±0.84 | 21.89±0.19 | 20.79±0.11 | 10.1 |
| BMAML | 59.89±0.68 | 37.04±0.85 | 44.35±0.81 | 48.27±0.68 | 33.10±0.60 | 54.93±0.59 | 20.00±0.00 | 20.00±0.00 | 12.6 |
| BMAML (chaser) | 56.45±0.67 | 32.05±0.75 | 37.73±0.77 | 40.47±0.56 | 27.90±0.51 | 52.25±0.59 | 20.00±0.00 | 20.00±0.00 | 13.5 |
| BTAML ($\alpha,\omega,\gamma,z$) | 52.94±0.76 | 36.46±1.28 | 51.39±0.74 | 52.06±0.74 | 20.00±0.00 | 20.00±0.00 | - | - | 14.6 |
| BTAML ($\alpha,\gamma,z$) | 54.89±0.75 | 37.32±1.33 | 52.92±0.75 | 53.51±0.74 | 20.00±0.00 | 20.00±0.00 | - | - | 13.5 |
| BTAML ($\alpha$) | 52.19±0.76 | 46.15±0.73 | 50.16±0.75 | 51.04±0.75 | 44.61±0.64 | 49.41±0.66 | - | - | 12.6 |

Table 15: Random-Shot (accuracy)

| | 5shot balanced | 1-9shot random | 3-7shot random | 4-6shot linear | 1-9shot linear | 1-9shot step 1minor | 1-9shot step 4minor | 1-21shot step 4minor | Rank |
|---|---|---|---|---|---|---|---|---|---|
| Baseline (1-NN) | 40.83±0.74 | 39.13±0.72 | 40.44±0.72 | 40.53±0.74 | 38.68±0.72 | 40.60±0.70 | 32.18±0.70 | 30.08±0.66 | 11.8 |
| Baseline (fine-tune) | 62.83±0.70 | 49.96±0.84 | 57.31±0.73 | 59.94±0.69 | 47.44±0.58 | 58.60±0.55 | 21.24±0.18 | 20.33±0.08 | 6.6 |
| Baseline++ | **66.74±0.65** | 49.06±0.89 | 57.44±0.77 | 61.41±0.68 | 46.46±0.60 | 61.07±0.54 | 21.13±0.20 | 20.28±0.09 | 6.5 |
| Matching Net | 59.58±0.69 | 54.79±0.74 | 58.24±0.69 | 58.79±0.69 | 53.47±0.69 | 58.19±0.65 | 39.20±0.68 | 33.11±0.62 | 5.1 |
| ProtoNet | 51.65±0.68 | 49.24±0.69 | 50.87±0.70 | 51.31±0.68 | 48.57±0.65 | 50.89±0.63 | 39.88±0.66 | 39.43±0.65 | 7.2 |
| ProtoNet (20-way) | 59.79±0.70 | 56.03±0.76 | 58.99±0.71 | 59.52±0.71 | 54.66±0.67 | 57.32±0.62 | 42.24±0.72 | 41.29±0.67 | 4.1 |
| Relation Net (CE) | 63.50±0.70 | **61.33±0.74** | **62.99±0.71** | **63.30±0.70** | **60.64±0.71** | **63.33±0.69** | **51.20±0.76** | **51.69±0.75** | **1.1** |
| DKT | 63.23±0.66 | 58.92±0.71 | 61.97±0.68 | 62.72±0.67 | 57.42±0.67 | 60.75±0.62 | 47.37±0.71 | 45.20±0.70 | 2.2 |
| SimpleShot | 61.99±0.71 | 50.41±0.88 | 56.77±0.73 | 59.35±0.71 | 48.19±0.60 | 57.57±0.59 | 21.08±0.16 | 20.67±0.12 | 7.1 |
| MAML | 50.79±0.67 | 43.14±0.71 | 47.75±0.68 | 49.25±0.67 | 41.49±0.58 | 48.32±0.61 | 23.88±0.28 | 20.28±0.07 | 11.2 |
| ProtoMAML | 54.78±0.72 | 46.09±0.78 | 51.61±0.74 | 53.12±0.72 | 44.04±0.65 | 49.69±0.66 | 22.20±0.22 | 21.10±0.14 | 9.0 |
| BMAML | 50.51±0.64 | 43.60±0.71 | 48.12±0.66 | 49.35±0.64 | 41.95±0.62 | 47.26±0.59 | 26.61±0.49 | 21.33±0.22 | 10.2 |
| BMAML (chaser) | 45.63±0.61 | 38.66±0.65 | 42.70±0.62 | 44.03±0.61 | 37.63±0.56 | 43.35±0.57 | 23.48±0.34 | 20.45±0.10 | 12.4 |
| BTAML ($\alpha,\omega,\gamma,z$) | 20.00±0.00 | 20.00±0.00 | 20.00±0.00 | 20.00±0.00 | 20.00±0.00 | 20.00±0.00 | - | - | 16.0 |
| BTAML ($\alpha,\gamma,z$) | 20.00±0.00 | 20.00±0.00 | 20.00±0.00 | 20.00±0.00 | 20.00±0.00 | 20.00±0.00 | - | - | 15.0 |
| BTAML ($\alpha$) | 52.66±0.77 | 46.85±0.74 | 51.13±0.76 | 52.10±0.77 | 45.08±0.65 | 50.12±0.67 | - | - | 10.2 |

Table 16: Random-Shot (ROS) (accuracy)

| | 5shot balanced | 1-9shot random | 3-7shot random | 4-6shot linear | 1-9shot linear | 1-9shot step 1minor | 1-9shot step 4minor | 1-21shot step 4minor | Rank |
|---|---|---|---|---|---|---|---|---|---|
| Baseline (1-NN) | 39.90±0.75 | 38.29±0.74 | 39.43±0.74 | 39.63±0.75 | 37.95±0.73 | 39.83±0.70 | 31.76±0.68 | 29.81±0.65 | 14.4 |
| Baseline (fine-tune) | 63.01±0.69 | 51.38±0.85 | 58.61±0.72 | 60.78±0.69 | 49.75±0.60 | 58.90±0.56 | 23.53±0.32 | 21.64±0.22 | 7.8 |
| Baseline++ | **66.25±0.66** | 59.53±0.77 | **64.36±0.68** | **65.27±0.67** | 57.27±0.64 | 62.31±0.56 | 36.63±0.70 | 29.80±0.60 | 4.0 |
| Matching Net | 61.75±0.70 | 59.53±0.73 | 61.13±0.70 | 61.50±0.69 | 59.07±0.72 | 61.92±0.69 | 51.53±0.76 | 51.86±0.75 | 3.1 |
| ProtoNet | 55.35±0.68 | 51.32±0.70 | 53.79±0.68 | 54.40±0.68 | 50.40±0.64 | 50.40±0.64 | 39.72±0.67 | 39.08±0.65 | 8.9 |
| ProtoNet (20-way) | 59.52±0.71 | 54.84±0.74 | 57.67±0.72 | 58.30±0.72 | 53.49±0.67 | 56.89±0.63 | 41.50±0.73 | 40.62±0.72 | 6.5 |
| Relation Net (CE) | 64.12±0.71 | **61.32±0.77** | 62.86±0.71 | 63.31±0.70 | **60.93±0.71** | **64.20±0.68** | 52.16±0.76 | 52.60±0.76 | **1.4** |
| DKT | 63.21±0.67 | 59.80±0.72 | 62.18±0.68 | 62.85±0.68 | 58.65±0.68 | 61.52±0.65 | 48.85±0.72 | 46.92±0.72 | 3.1 |
| SimpleShot | 59.52±0.73 | 47.63±0.80 | 53.35±0.75 | 55.39±0.71 | 46.04±0.63 | 55.67±0.60 | 22.20±0.24 | 21.60±0.20 | 11.0 |
| MAML | 59.10±0.70 | 54.09±0.73 | 57.61±0.70 | 58.30±0.69 | 52.81±0.66 | 56.71±0.63 | 39.24±0.69 | 37.23±0.68 | 7.5 |
| ProtoMAML | 54.50±0.90 | 48.04±0.84 | 52.59±0.84 | 53.96±0.65 | 46.59±0.80 | 45.06±0.81 | 31.19±0.60 | 29.23±0.55 | 12.2 |
| BMAML | 58.98±0.68 | 55.99±0.73 | 57.70±0.71 | 58.24±0.69 | 55.23±0.72 | 57.54±0.71 | 48.22±0.74 | 48.38±0.74 | 5.8 |
| BMAML (chaser) | 53.52±0.64 | 50.90±0.67 | 52.62±0.66 | 52.87±0.64 | 50.46±0.68 | 52.20±0.67 | 44.48±0.73 | 44.61±0.72 | 9.5 |
| BTAML ($\alpha,\omega,\gamma,z$) | 47.75±0.73 | 45.57±0.74 | 47.17±0.74 | 47.47±0.74 | 45.04±0.73 | 47.44±0.73 | - | - | 15.1 |
| BTAML ($\alpha,\gamma,z$) | 53.34±0.75 | 50.84±0.78 | 52.75±0.76 | 52.84±0.76 | 49.97±0.75 | 53.33±0.74 | - | - | 12.4 |
| BTAML ($\alpha$) | 51.59±0.77 | 49.00±0.78 | 50.90±0.77 | 51.23±0.77 | 48.62±0.75 | 51.68±0.74 | - | - | 13.4 |

Table 17: Random-Shot (ROS+) (accuracy)

| | 5shot balanced | 1-9shot random | 3-7shot random | 4-6shot linear | 1-9shot linear | 1-9shot step 1minor | 1-9shot step 4minor | 1-21shot step 4minor | Rank |
|---|---|---|---|---|---|---|---|---|---|
| Baseline (1-NN) | 28.83±0.53 | 29.00±0.56 | 28.49±0.53 | 28.93±0.54 | 33.48±0.61 | 34.74±0.63 | 33.89±0.64 | 34.75±0.65 | 15.1 |
| Baseline (fine-tune) | 60.46±0.70 | 53.33±0.79 | 57.53±0.74 | 59.44±0.72 | 56.12±0.69 | 61.81±0.64 | 42.21±0.70 | 39.00±0.69 | 7.1 |
| Baseline++ | **63.72±0.68** | 55.60±0.80 | 60.47±0.73 | **62.31±0.68** | 58.41±0.73 | **64.69±0.64** | 45.79±0.74 | 43.05±0.73 | 3.5 |
| Matching Net | 60.05±0.68 | 57.28±0.70 | 59.33±0.68 | 59.59±0.67 | 57.24±0.69 | 60.36±0.67 | **49.00±0.74** | **48.93±0.73** | 4.8 |
| ProtoNet | 60.05±0.71 | 57.35±0.74 | 59.94±0.70 | 59.98±0.69 | 56.89±0.68 | 60.57±0.66 | 46.23±0.74 | 45.83±0.73 | 5.4 |
| ProtoNet (20-way) | 61.21±0.72 | **59.89±0.75** | **61.73±0.71** | 61.41±0.71 | **58.58±0.69** | 62.84±0.66 | 47.02±0.74 | 46.77±0.72 | **2.1** |
| Relation Net (CE) | 60.48±0.71 | 58.20±0.73 | 60.01±0.71 | 60.50±0.70 | 57.99±0.72 | 61.44±0.69 | 48.64±0.76 | 48.35±0.75 | 3.1 |
| DKT | 61.16±0.67 | 58.18±0.72 | 60.08±0.68 | 60.17±0.67 | 57.71±0.70 | 61.87±0.66 | 48.22±0.72 | 47.01±0.71 | 3.2 |
| SimpleShot | 59.44±0.69 | 49.95±0.83 | 55.57±0.74 | 57.89±0.70 | 51.55±0.69 | 59.84±0.60 | 34.70±0.67 | 32.64±0.64 | 9.8 |
| MAML | 54.60±0.72 | 51.04±0.76 | 53.57±0.72 | 54.41±0.70 | 51.25±0.74 | 56.35±0.71 | 41.99±0.77 | 41.84±0.75 | 9.8 |
| ProtoMAML | 50.38±0.75 | 45.84±0.78 | 48.41±0.74 | 49.29±0.75 | 46.01±0.75 | 50.38±0.74 | 39.88±0.76 | 39.24±0.75 | 13.1 |
| BMAML | 56.52±0.69 | 53.11±0.75 | 55.81±0.71 | 56.07±0.68 | 53.23±0.71 | 56.92±0.68 | 46.02±0.72 | 46.03±0.71 | 7.8 |
| BMAML (chaser) | 50.49±0.73 | 50.65±0.71 | 52.71±0.68 | 52.92±0.68 | 50.68±0.70 | 52.29±0.67 | 43.59±0.71 | 43.72±0.70 | 10.2 |
| BTAML ($\alpha,\omega,\gamma,z$) | 47.97±0.72 | 44.28±0.74 | 46.29±0.73 | 46.85±0.74 | 43.60±0.73 | 47.05±0.72 | - | - | 15.2 |
| BTAML ($\alpha,\gamma,z$) | 52.66±0.76 | 48.82±0.78 | 50.88±0.77 | 51.54±0.75 | 47.79±0.76 | 51.97±0.75 | - | - | 12.6 |
| BTAML ($\alpha$) | 51.80±0.75 | 47.69±0.76 | 49.84±0.76 | 50.48±0.76 | 46.94±0.76 | 50.91±0.73 | - | - | 13.1 |

### Table 18: Standard (F1)

| | 5shot balanced | 1-9shot random | 3-7shot random | 4-6shot linear | 1-9shot linear | 1-9shot step 1minor | 1-9shot step 4minor | 1-21shot step 4minor | Rank |
|---|---|---|---|---|---|---|---|---|---|
| Baseline (1-NN) | $34.73_{\pm1.71}$ | $32.37_{\pm1.78}$ | $34.47_{\pm1.74}$ | $34.64_{\pm1.71}$ | $31.59_{\pm1.62}$ | $33.90_{\pm1.57}$ | $23.13_{\pm1.61}$ | $20.38_{\pm1.51}$ | 11.5 |
| Baseline (fine-tune) | $61.77_{\pm1.36}$ | $43.18_{\pm2.16}$ | $54.10_{\pm1.73}$ | $57.82_{\pm1.48}$ | $39.19_{\pm1.31}$ | $52.13_{\pm1.05}$ | $8.87_{\pm0.58}$ | $7.27_{\pm0.29}$ | 7.8 |
| Baseline++ | $65.06_{\pm1.38}$ | $41.17_{\pm2.35}$ | $52.31_{\pm2.03}$ | $57.84_{\pm1.65}$ | $36.55_{\pm1.41}$ | $54.29_{\pm1.04}$ | $8.59_{\pm0.62}$ | $7.16_{\pm0.29}$ | 7.6 |
| Matching Net | $60.92_{\pm1.41}$ | $53.76_{\pm1.79}$ | $58.87_{\pm1.52}$ | $59.68_{\pm1.46}$ | $49.70_{\pm1.54}$ | $54.20_{\pm1.33}$ | $34.56_{\pm1.86}$ | $26.59_{\pm1.79}$ | 4.2 |
| ProtoNet | $63.76_{\pm1.35}$ | $50.68_{\pm2.11}$ | $60.13_{\pm1.54}$ | $62.09_{\pm1.41}$ | $46.94_{\pm1.33}$ | $53.47_{\pm1.10}$ | $17.26_{\pm1.39}$ | $15.25_{\pm1.26}$ | 4.1 |
| ProtoNet (20-way) | $\mathbf{65.15}_{\pm1.34}$ | $48.37_{\pm2.23}$ | $59.82_{\pm1.64}$ | $62.68_{\pm1.45}$ | $45.26_{\pm1.36}$ | $54.17_{\pm1.07}$ | $12.10_{\pm1.01}$ | $10.74_{\pm0.87}$ | 4.1 |
| Relation Net (CE) | $64.00_{\pm1.36}$ | $55.27_{\pm1.91}$ | $\mathbf{62.32}_{\pm1.46}$ | $\mathbf{63.19}_{\pm1.40}$ | $51.39_{\pm1.42}$ | $55.12_{\pm1.25}$ | $30.63_{\pm1.81}$ | $28.58_{\pm1.80}$ | **2.0** |
| DKT | $61.51_{\pm1.40}$ | $\mathbf{55.65}_{\pm1.74}$ | $59.71_{\pm1.50}$ | $60.73_{\pm1.43}$ | $\mathbf{53.49}_{\pm1.53}$ | $\mathbf{56.73}_{\pm1.38}$ | $\mathbf{42.88}_{\pm1.78}$ | $\mathbf{40.12}_{\pm1.81}$ | 2.5 |
| SimpleShot | $63.19_{\pm1.32}$ | $41.90_{\pm2.29}$ | $53.10_{\pm1.83}$ | $57.83_{\pm1.49}$ | $38.28_{\pm1.31}$ | $52.79_{\pm1.05}$ | $7.07_{\pm0.23}$ | $6.91_{\pm0.17}$ | 8.5 |
| MAML | $61.22_{\pm1.36}$ | $45.22_{\pm2.05}$ | $55.65_{\pm1.56}$ | $58.56_{\pm1.41}$ | $40.94_{\pm1.24}$ | $50.48_{\pm1.08}$ | $11.66_{\pm0.91}$ | $7.26_{\pm0.26}$ | 7.1 |
| ProtoMAML | $59.13_{\pm1.45}$ | $39.41_{\pm2.17}$ | $51.06_{\pm1.82}$ | $54.84_{\pm1.62}$ | $35.58_{\pm1.73}$ | $36.01_{\pm2.03}$ | $10.04_{\pm0.67}$ | $8.14_{\pm0.41}$ | 10.1 |
| BMAML | $56.87_{\pm1.68}$ | $26.26_{\pm2.23}$ | $35.54_{\pm2.29}$ | $40.34_{\pm1.82}$ | $21.69_{\pm1.36}$ | $47.76_{\pm1.27}$ | $6.67_{\pm0.00}$ | $6.67_{\pm0.00}$ | 12.9 |
| BMAML (chaser) | $53.26_{\pm1.66}$ | $20.50_{\pm2.00}$ | $27.48_{\pm2.15}$ | $30.17_{\pm1.45}$ | $15.93_{\pm1.10}$ | $45.20_{\pm1.25}$ | $6.67_{\pm0.00}$ | $6.67_{\pm0.00}$ | 13.8 |
| BTAML $(\alpha,\omega,\gamma,z)$ | $52.30_{\pm1.43}$ | $29.27_{\pm2.06}$ | $50.06_{\pm1.50}$ | $51.09_{\pm1.43}$ | $6.67_{\pm0.00}$ | $6.67_{\pm0.00}$ | - | - | 14.5 |
| BTAML $(\alpha,\gamma,z)$ | $54.52_{\pm1.39}$ | $30.19_{\pm2.11}$ | $51.80_{\pm1.46}$ | $52.79_{\pm1.39}$ | $6.67_{\pm0.00}$ | $6.67_{\pm0.00}$ | - | - | 13.2 |
| BTAML $(\alpha)$ | $51.85_{\pm1.41}$ | $42.61_{\pm1.73}$ | $49.09_{\pm1.46}$ | $50.35_{\pm1.40}$ | $39.51_{\pm1.26}$ | $44.33_{\pm1.16}$ | - | - | 12.0 |

### Table 19: Random-Shot (F1)

| | 5shot balanced | 1-9shot random | 3-7shot random | 4-6shot linear | 1-9shot linear | 1-9shot step 1minor | 1-9shot step 4minor | 1-21shot step 4minor | Rank |
|---|---|---|---|---|---|---|---|---|---|
| Baseline (1-NN) | $36.16_{\pm1.73}$ | $33.45_{\pm1.80}$ | $35.59_{\pm1.74}$ | $35.71_{\pm1.73}$ | $32.63_{\pm1.65}$ | $35.07_{\pm1.58}$ | $23.80_{\pm1.62}$ | $20.75_{\pm1.52}$ | 11.8 |
| Baseline (fine-tune) | $61.92_{\pm1.36}$ | $43.39_{\pm2.16}$ | $54.26_{\pm1.72}$ | $57.99_{\pm1.47}$ | $39.23_{\pm1.29}$ | $52.29_{\pm1.04}$ | $8.84_{\pm0.57}$ | $7.27_{\pm0.29}$ | 7.1 |
| Baseline++ | $\mathbf{65.34}_{\pm1.39}$ | $40.82_{\pm2.36}$ | $52.59_{\pm2.04}$ | $58.07_{\pm1.66}$ | $36.80_{\pm1.41}$ | $54.37_{\pm1.05}$ | $8.51_{\pm0.61}$ | $7.14_{\pm0.30}$ | 7.5 |
| Matching Net | $57.51_{\pm1.53}$ | $50.70_{\pm1.86}$ | $55.55_{\pm1.63}$ | $56.41_{\pm1.56}$ | $48.84_{\pm1.70}$ | $53.99_{\pm1.49}$ | $32.62_{\pm1.85}$ | $24.83_{\pm1.77}$ | 5.0 |
| ProtoNet | $50.65_{\pm1.45}$ | $46.87_{\pm1.67}$ | $49.70_{\pm1.49}$ | $50.26_{\pm1.45}$ | $45.44_{\pm1.48}$ | $47.66_{\pm1.41}$ | $35.17_{\pm1.71}$ | $34.35_{\pm1.71}$ | 6.9 |
| ProtoNet (20-way) | $59.13_{\pm1.43}$ | $53.45_{\pm1.79}$ | $58.12_{\pm1.47}$ | $58.75_{\pm1.44}$ | $50.95_{\pm1.50}$ | $53.57_{\pm1.39}$ | $37.07_{\pm1.16}$ | $35.76_{\pm1.85}$ | 3.6 |
| Relation Net (CE) | $62.54_{\pm1.43}$ | $\mathbf{60.18}_{\pm1.51}$ | $\mathbf{62.01}_{\pm1.44}$ | $\mathbf{62.35}_{\pm1.42}$ | $\mathbf{59.34}_{\pm1.48}$ | $\mathbf{62.15}_{\pm1.42}$ | $\mathbf{49.25}_{\pm1.64}$ | $\mathbf{49.67}_{\pm1.64}$ | **1.1** |
| DKT | $61.80_{\pm1.41}$ | $55.88_{\pm1.74}$ | $60.15_{\pm1.50}$ | $61.11_{\pm1.44}$ | $53.45_{\pm1.54}$ | $56.74_{\pm1.38}$ | $42.75_{\pm1.80}$ | $39.91_{\pm1.83}$ | 2.2 |
| SimpleShot | $61.44_{\pm1.34}$ | $44.37_{\pm2.18}$ | $54.13_{\pm1.71}$ | $57.82_{\pm1.44}$ | $40.59_{\pm1.31}$ | $51.59_{\pm1.09}$ | $8.56_{\pm0.53}$ | $7.86_{\pm0.41}$ | 7.1 |
| MAML | $48.61_{\pm1.52}$ | $36.44_{\pm1.97}$ | $43.85_{\pm1.73}$ | $46.17_{\pm1.57}$ | $33.49_{\pm1.32}$ | $41.91_{\pm1.21}$ | $13.46_{\pm0.97}$ | $7.20_{\pm0.26}$ | 10.8 |
| ProtoMAML | $54.12_{\pm1.41}$ | $40.52_{\pm1.98}$ | $49.38_{\pm1.62}$ | $51.70_{\pm1.47}$ | $37.19_{\pm1.37}$ | $43.34_{\pm1.38}$ | $10.60_{\pm0.74}$ | $8.72_{\pm0.51}$ | 9.1 |
| BMAML | $46.11_{\pm1.77}$ | $35.90_{\pm2.07}$ | $42.55_{\pm1.92}$ | $44.29_{\pm1.80}$ | $33.23_{\pm1.58}$ | $39.84_{\pm1.44}$ | $16.07_{\pm1.39}$ | $8.76_{\pm0.67}$ | 10.8 |
| BMAML (chaser) | $41.04_{\pm1.69}$ | $30.50_{\pm1.94}$ | $36.53_{\pm1.84}$ | $38.57_{\pm1.69}$ | $28.44_{\pm1.39}$ | $35.99_{\pm1.36}$ | $12.01_{\pm1.01}$ | $7.46_{\pm0.35}$ | 12.4 |
| BTAML $(\alpha,\omega,\gamma,z)$ | $6.67_{\pm0.00}$ | $6.67_{\pm0.00}$ | $6.67_{\pm0.00}$ | $6.67_{\pm0.00}$ | $6.67_{\pm0.00}$ | $6.67_{\pm0.00}$ | - | - | 16.0 |
| BTAML $(\alpha,\gamma,z)$ | $6.67_{\pm0.00}$ | $6.67_{\pm0.00}$ | $6.67_{\pm0.00}$ | $6.67_{\pm0.00}$ | $6.67_{\pm0.00}$ | $6.67_{\pm0.00}$ | - | - | 15.0 |
| BTAML $(\alpha)$ | $52.37_{\pm1.41}$ | $43.46_{\pm1.72}$ | $50.09_{\pm1.47}$ | $51.49_{\pm1.41}$ | $40.09_{\pm1.27}$ | $45.14_{\pm1.17}$ | - | - | 9.6 |

### Table 20: Random-Shot (ROS) (F1)

| | 5shot balanced | 1-9shot random | 3-7shot random | 4-6shot linear | 1-9shot linear | 1-9shot step 1minor | 1-9shot step 4minor | 1-21shot step 4minor | Rank |
|---|---|---|---|---|---|---|---|---|---|
| Baseline (1-NN) | $34.95_{\pm1.70}$ | $32.32_{\pm1.78}$ | $34.26_{\pm1.71}$ | $34.54_{\pm1.70}$ | $31.70_{\pm1.62}$ | $34.08_{\pm1.56}$ | $23.11_{\pm1.61}$ | $20.31_{\pm1.51}$ | 14.8 |
| Baseline (fine-tune) | $62.15_{\pm1.35}$ | $45.48_{\pm2.11}$ | $56.16_{\pm1.63}$ | $59.16_{\pm1.43}$ | $42.44_{\pm1.33}$ | $52.77_{\pm1.05}$ | $12.44_{\pm0.99}$ | $9.46_{\pm0.68}$ | 8.9 |
| Baseline++ | $\mathbf{64.83}_{\pm1.39}$ | $55.35_{\pm1.95}$ | $62.44_{\pm1.51}$ | $63.64_{\pm1.43}$ | $51.73_{\pm1.50}$ | $56.90_{\pm1.25}$ | $29.57_{\pm1.88}$ | $20.73_{\pm1.65}$ | 4.2 |
| Matching Net | $60.35_{\pm1.44}$ | $57.85_{\pm1.54}$ | $59.65_{\pm1.47}$ | $60.05_{\pm1.45}$ | $57.28_{\pm1.51}$ | $60.21_{\pm1.44}$ | $49.27_{\pm1.65}$ | $49.51_{\pm1.64}$ | 2.9 |
| ProtoNet | $54.56_{\pm1.41}$ | $48.61_{\pm1.70}$ | $52.55_{\pm1.49}$ | $53.38_{\pm1.44}$ | $46.74_{\pm1.47}$ | $50.01_{\pm1.36}$ | $34.41_{\pm1.73}$ | $33.41_{\pm1.73}$ | 9.1 |
| ProtoNet (20-way) | $58.87_{\pm1.43}$ | $52.07_{\pm1.80}$ | $56.54_{\pm1.53}$ | $57.40_{\pm1.48}$ | $49.73_{\pm1.52}$ | $53.03_{\pm1.39}$ | $36.20_{\pm1.82}$ | $34.92_{\pm1.84}$ | 6.0 |
| Relation Net (CE) | $63.16_{\pm1.42}$ | $\mathbf{60.08}_{\pm1.54}$ | $61.81_{\pm1.47}$ | $62.30_{\pm1.43}$ | $\mathbf{59.56}_{\pm1.49}$ | $\mathbf{62.95}_{\pm1.41}$ | $50.27_{\pm1.65}$ | $50.62_{\pm1.64}$ | **1.4** |
| DKT | $61.83_{\pm1.40}$ | $57.22_{\pm1.67}$ | $60.55_{\pm1.46}$ | $61.36_{\pm1.41}$ | $55.36_{\pm1.53}$ | $58.19_{\pm1.42}$ | $44.83_{\pm1.75}$ | $42.28_{\pm1.79}$ | 3.2 |
| SimpleShot | $58.93_{\pm1.38}$ | $41.58_{\pm2.09}$ | $50.55_{\pm1.72}$ | $53.33_{\pm1.52}$ | $38.84_{\pm1.35}$ | $49.96_{\pm1.11}$ | $10.36_{\pm0.79}$ | $9.40_{\pm0.68}$ | 11.9 |
| MAML | $58.35_{\pm1.38}$ | $51.10_{\pm1.74}$ | $56.34_{\pm1.46}$ | $57.32_{\pm1.40}$ | $48.80_{\pm1.44}$ | $52.56_{\pm1.30}$ | $34.05_{\pm1.72}$ | $31.26_{\pm1.72}$ | 7.5 |
| ProtoMAML | $52.10_{\pm1.81}$ | $42.91_{\pm2.03}$ | $49.85_{\pm1.80}$ | $51.54_{\pm1.77}$ | $40.36_{\pm1.74}$ | $36.75_{\pm2.04}$ | $23.18_{\pm1.55}$ | $20.46_{\pm1.46}$ | 12.5 |
| BMAML | $55.76_{\pm1.71}$ | $52.63_{\pm1.77}$ | $54.37_{\pm1.74}$ | $55.00_{\pm1.73}$ | $51.83_{\pm1.78}$ | $54.31_{\pm1.74}$ | $44.72_{\pm1.79}$ | $44.87_{\pm1.79}$ | 5.8 |
| BMAML (chaser) | $49.72_{\pm1.71}$ | $47.17_{\pm1.72}$ | $48.88_{\pm1.71}$ | $49.14_{\pm1.70}$ | $46.75_{\pm1.73}$ | $48.55_{\pm1.70}$ | $40.92_{\pm1.74}$ | $41.06_{\pm1.73}$ | 10.8 |
| BTAML $(\alpha,\omega,\gamma,z)$ | $45.92_{\pm1.55}$ | $43.88_{\pm1.55}$ | $45.45_{\pm1.54}$ | $45.71_{\pm1.54}$ | $43.35_{\pm1.52}$ | $45.48_{\pm1.54}$ | - | - | 14.5 |
| BTAML $(\alpha,\gamma,z)$ | $52.79_{\pm1.42}$ | $49.97_{\pm1.50}$ | $52.17_{\pm1.44}$ | $52.23_{\pm1.43}$ | $48.83_{\pm1.45}$ | $52.06_{\pm1.43}$ | - | - | 10.8 |
| BTAML $(\alpha)$ | $51.27_{\pm1.41}$ | $48.29_{\pm1.48}$ | $50.51_{\pm1.42}$ | $50.86_{\pm1.41}$ | $47.62_{\pm1.42}$ | $50.53_{\pm1.40}$ | - | - | 11.9 |

### Table 21: Random-Shot (ROS+) (F1)

| | 5shot balanced | 1-9shot random | 3-7shot random | 4-6shot linear | 1-9shot linear | 1-9shot step 1minor | 1-9shot step 4minor | 1-21shot step 4minor | Rank |
|---|---|---|---|---|---|---|---|---|---|
| Baseline (1-NN) | $20.25_{\pm1.43}$ | $20.13_{\pm1.46}$ | $19.70_{\pm1.42}$ | $20.38_{\pm1.44}$ | $26.00_{\pm1.48}$ | $27.86_{\pm1.54}$ | $26.61_{\pm1.44}$ | $27.97_{\pm1.47}$ | 15.1 |
| Baseline (fine-tune) | $59.51_{\pm1.40}$ | $49.43_{\pm1.89}$ | $55.84_{\pm1.57}$ | $58.32_{\pm1.43}$ | $53.47_{\pm1.49}$ | $59.00_{\pm1.37}$ | $36.70_{\pm1.61}$ | $32.47_{\pm1.61}$ | 7.1 |
| Baseline++ | $\mathbf{61.95}_{\pm1.46}$ | $51.37_{\pm1.95}$ | $58.12_{\pm1.65}$ | $60.26_{\pm1.50}$ | $55.34_{\pm1.63}$ | $\mathbf{61.95}_{\pm1.43}$ | $40.69_{\pm1.70}$ | $36.90_{\pm1.72}$ | 4.1 |
| Matching Net | $58.50_{\pm1.46}$ | $54.58_{\pm1.67}$ | $57.50_{\pm1.52}$ | $57.84_{\pm1.47}$ | $54.93_{\pm1.54}$ | $58.18_{\pm1.47}$ | $45.52_{\pm1.69}$ | $44.99_{\pm1.69}$ | 5.1 |
| ProtoNet | $59.30_{\pm1.41}$ | $55.62_{\pm1.63}$ | $59.10_{\pm1.42}$ | $59.22_{\pm1.41}$ | $54.66_{\pm1.47}$ | $58.59_{\pm1.40}$ | $43.40_{\pm1.66}$ | $42.31_{\pm1.69}$ | 4.5 |
| ProtoNet (20-way) | $60.27_{\pm1.45}$ | $\mathbf{58.69}_{\pm1.54}$ | $\mathbf{60.84}_{\pm1.43}$ | $\mathbf{60.56}_{\pm1.44}$ | $56.08_{\pm1.47}$ | $60.73_{\pm1.40}$ | $43.75_{\pm1.66}$ | $42.93_{\pm1.67}$ | **2.0** |
| Relation Net (CE) | $59.41_{\pm1.45}$ | $56.54_{\pm1.59}$ | $58.76_{\pm1.48}$ | $59.35_{\pm1.44}$ | $\mathbf{56.15}_{\pm1.53}$ | $59.91_{\pm1.46}$ | $\mathbf{46.31}_{\pm1.66}$ | $\mathbf{45.70}_{\pm1.66}$ | 2.2 |
| DKT | $59.15_{\pm1.46}$ | $55.52_{\pm1.68}$ | $57.97_{\pm1.54}$ | $57.88_{\pm1.49}$ | $55.12_{\pm1.54}$ | $59.44_{\pm1.42}$ | $44.27_{\pm1.70}$ | $42.37_{\pm1.69}$ | 4.5 |
| SimpleShot | $58.03_{\pm1.39}$ | $44.91_{\pm2.06}$ | $53.63_{\pm1.62}$ | $56.39_{\pm1.43}$ | $46.98_{\pm1.48}$ | $55.49_{\pm1.26}$ | $27.67_{\pm1.57}$ | $24.72_{\pm1.52}$ | 10.2 |
| MAML | $53.90_{\pm1.40}$ | $49.73_{\pm1.55}$ | $52.64_{\pm1.44}$ | $53.60_{\pm1.40}$ | $50.10_{\pm1.47}$ | $55.21_{\pm1.40}$ | $40.22_{\pm1.53}$ | $39.78_{\pm1.52}$ | 8.4 |
| ProtoMAML | $48.51_{\pm1.58}$ | $42.94_{\pm1.71}$ | $45.99_{\pm1.62}$ | $46.99_{\pm1.62}$ | $43.11_{\pm1.63}$ | $47.51_{\pm1.63}$ | $37.76_{\pm1.56}$ | $36.63_{\pm1.57}$ | 12.9 |
| BMAML | $52.78_{\pm1.75}$ | $48.41_{\pm1.93}$ | $51.97_{\pm1.81}$ | $52.28_{\pm1.75}$ | $53.26_{\pm1.73}$ | $49.12_{\pm1.81}$ | $41.32_{\pm1.76}$ | $41.21_{\pm1.74}$ | 8.8 |
| BMAML (chaser) | $46.35_{\pm1.83}$ | $46.48_{\pm1.79}$ | $49.07_{\pm1.73}$ | $49.46_{\pm1.71}$ | $47.17_{\pm1.69}$ | $48.88_{\pm1.75}$ | $39.64_{\pm1.68}$ | $39.89_{\pm1.66}$ | 11.4 |
| BTAML $(\alpha,\omega,\gamma,z)$ | $46.32_{\pm1.52}$ | $42.63_{\pm1.54}$ | $44.64_{\pm1.52}$ | $45.14_{\pm1.54}$ | $41.97_{\pm1.52}$ | $45.32_{\pm1.53}$ | - | - | 15.2 |
| BTAML $(\alpha,\gamma,z)$ | $52.06_{\pm1.43}$ | $48.10_{\pm1.49}$ | $50.27_{\pm1.44}$ | $50.90_{\pm1.43}$ | $47.06_{\pm1.45}$ | $51.12_{\pm1.43}$ | - | - | 11.9 |
| BTAML $(\alpha)$ | $51.41_{\pm1.40}$ | $47.16_{\pm1.45}$ | $49.41_{\pm1.41}$ | $50.05_{\pm1.41}$ | $46.35_{\pm1.41}$ | $50.21_{\pm1.39}$ | - | - | 12.5 |

Table 22: Full results for Table 1.

| $\mathcal{I}$-Distribution | Imbalanced Mini-ImageNet | | | | | | Imbalanced Mini-ImageNet $\to$ CUB-200-2011 | | | | | |
|---|---|---|---|---|---|---|---|---|---|---|---|---|
| | balanced | linear | random | step | | | balanced | linear | random | step | | |
| Max. # samples, $D_{train}^{K_{max}}$ | 300 | 570 | 570 | 462 | 570 | 600 | 300 | 570 | 570 | 462 | 570 | 600 |
| Min. # samples, $D_{train}^{K_{min}}$ | 300 | 30 | 30 | 30 | 30 | 120 | 300 | 30 | 30 | 30 | 30 | 120 |
| # Minority, $D_{train}^{N_{min}}$ | - | - | - | 24 | 32 | 40 | - | - | - | 24 | 32 | 40 |
| Baseline (1-NN) | $42.69_{\pm0.66}$ | $43.42_{\pm0.68}$ | $42.15_{\pm0.66}$ | $39.02_{\pm0.63}$ | $41.45_{\pm0.65}$ | $41.71_{\pm0.66}$ | $43.21_{\pm0.68}$ | $43.42_{\pm0.69}$ | $43.39_{\pm0.69}$ | $42.03_{\pm0.65}$ | $42.19_{\pm0.66}$ | $42.99_{\pm0.66}$ |
| Baseline (fine-tune) | $51.26_{\pm0.70}$ | $50.13_{\pm0.69}$ | $54.16_{\pm0.72}$ | $51.54_{\pm0.71}$ | $52.47_{\pm0.70}$ | $56.48_{\pm0.71}$ | $53.19_{\pm0.71}$ | $51.95_{\pm0.72}$ | $53.52_{\pm0.72}$ | $52.47_{\pm0.71}$ | $52.68_{\pm0.70}$ | $53.72_{\pm0.71}$ |
| Baseline++ | $48.44_{\pm0.65}$ | $47.18_{\pm0.64}$ | $51.47_{\pm0.67}$ | $49.01_{\pm0.66}$ | $51.88_{\pm0.69}$ | $50.54_{\pm0.67}$ | $49.38_{\pm0.69}$ | $46.83_{\pm0.67}$ | $50.48_{\pm0.67}$ | $48.12_{\pm0.67}$ | $48.42_{\pm0.67}$ | $49.42_{\pm0.69}$ |
| Matching Net | $58.26_{\pm0.68}$ | $58.24_{\pm0.69}$ | $58.45_{\pm0.68}$ | $57.06_{\pm0.69}$ | $56.53_{\pm0.69}$ | $58.18_{\pm0.70}$ | $50.92_{\pm0.74}$ | $51.32_{\pm0.73}$ | $50.77_{\pm0.76}$ | $50.82_{\pm0.73}$ | $50.51_{\pm0.73}$ | $49.95_{\pm0.74}$ |
| ProtoNet | $60.65_{\pm0.70}$ | $59.17_{\pm0.68}$ | $60.16_{\pm0.70}$ | $\mathbf{59.34}_{\pm\mathbf{0.70}}$ | $58.69_{\pm0.72}$ | $60.07_{\pm0.71}$ | $52.86_{\pm0.73}$ | $51.85_{\pm0.72}$ | $52.06_{\pm0.71}$ | $53.32_{\pm0.65}$ | $52.42_{\pm0.71}$ | $51.34_{\pm0.71}$ |
| ProtoNet (20-way) | $60.91_{\pm0.70}$ | $60.64_{\pm0.70}$ | $60.37_{\pm0.70}$ | $59.02_{\pm0.71}$ | $58.83_{\pm0.70}$ | $60.87_{\pm0.70}$ | $52.80_{\pm0.72}$ | $52.60_{\pm0.74}$ | $52.31_{\pm0.73}$ | $51.15_{\pm0.72}$ | $51.33_{\pm0.72}$ | $52.61_{\pm0.72}$ |
| Relation Net (CE) | $\mathbf{62.78}_{\pm\mathbf{0.70}}$ | $\mathbf{61.39}_{\pm\mathbf{0.69}}$ | $\mathbf{62.35}_{\pm\mathbf{0.70}}$ | $58.80_{\pm0.71}$ | $57.93_{\pm0.72}$ | $\mathbf{61.35}_{\pm\mathbf{0.69}}$ | $54.32_{\pm0.68}$ | $\mathbf{54.41}_{\pm\mathbf{0.71}}$ | $52.13_{\pm0.65}$ | $51.30_{\pm0.65}$ | $49.90_{\pm0.62}$ | $52.76_{\pm0.67}$ |
| DKT | $58.09_{\pm0.69}$ | $57.59_{\pm0.68}$ | $57.81_{\pm0.67}$ | $56.99_{\pm0.67}$ | $55.91_{\pm0.67}$ | $57.64_{\pm0.68}$ | $\mathbf{54.62}_{\pm\mathbf{0.71}}$ | $54.19_{\pm0.71}$ | $54.86_{\pm0.72}$ | $\mathbf{53.95}_{\pm\mathbf{0.71}}$ | $\mathbf{54.44}_{\pm\mathbf{0.71}}$ | $\mathbf{54.05}_{\pm\mathbf{0.72}}$ |
| SimpleShot | $59.55_{\pm0.72}$ | $59.78_{\pm0.71}$ | $58.74_{\pm0.72}$ | $58.86_{\pm0.72}$ | $\mathbf{58.89}_{\pm\mathbf{0.71}}$ | $58.60_{\pm0.71}$ | $53.16_{\pm0.71}$ | $53.46_{\pm0.71}$ | $52.88_{\pm0.72}$ | $52.90_{\pm0.70}$ | $52.87_{\pm0.72}$ | $52.57_{\pm0.72}$ |
| MAML | $54.43_{\pm0.69}$ | $55.14_{\pm0.72}$ | $54.97_{\pm0.73}$ | $55.37_{\pm0.72}$ | $54.30_{\pm0.70}$ | $55.65_{\pm0.72}$ | $53.46_{\pm0.67}$ | $53.26_{\pm0.70}$ | $\mathbf{55.14}_{\pm\mathbf{0.67}}$ | $53.65_{\pm0.69}$ | $53.96_{\pm0.69}$ | $53.11_{\pm0.70}$ |
| ProtoMAML | $51.31_{\pm0.72}$ | $54.57_{\pm0.69}$ | $45.94_{\pm0.73}$ | $54.39_{\pm0.71}$ | $53.56_{\pm0.71}$ | $54.60_{\pm0.69}$ | $48.52_{\pm0.72}$ | $51.25_{\pm0.69}$ | $45.27_{\pm0.70}$ | $51.55_{\pm0.68}$ | $51.64_{\pm0.67}$ | $51.29_{\pm0.68}$ |

