# OpenReview forum: "Class Imbalance in Few-Shot Learning"
_ICLR.cc/2021/Conference — Reject_

### Official Review · AnonReviewer4 · 2020-10-22
**This paper conducts extensive comparison experiments to study the effect of class-imbalance for many few-shot approaches.**

**Rating:** 5
**Confidence:** 3

**Review:**

This paper conducts extensive comparison experiments to study the effect of class-imbalance for many few-shot approaches. A detailed study of few-shot class-imbalance along three axes: dataset vs. support set imbalance, effect of different imbalance distributions (linear, step, random), and effect of rebalancing techniques, are presented. Also, this paper is clearly written and easy to understand.

1. Though eleven few-shot approaches are considered, some strong baselines are missing, such as [1];
2. In the contribution part, this paper declares "compare over 10 state-of-the-art few-shot learning methods using backbones of different depths on multiple datasets", however, "backbones of different depths" is commonly used in few-shot learning literature. Therefore it cannot reflect much contribution of this paper;
3. Some related work is not discussed [2,3]. For instnace, prior work [2] discusses the effect of different value of $k$ in meta-training and meta-testing, which is pretty much similar to the concept "imbalance" studied in this paper;
4. Overall, the contribution of this paper is somewhat limited. Apart from conducting extensive experiments, more informative observations and conclusions should be made.

[1] A Baseline for Few-Shot Image Classification. ICLR 2020.
[2] A Theoretical Analysis of the Number of Shots in Few-Shot Learning. ICLR 2020.
[3] Learning to Stop While Learning to Predict. ICML 2020.

---

> ### Author Response · Authors · 2020-11-17
> **Author Response 1 to AnonReviewer4**
>
> We thank the reviewer for the positive comments and constructive suggestions. We respond to your feedback below:
> 1. “[...] some strong baselines are missing, such as [1]”
>    * A. We thank the reviewer for drawing our attention to this work; it is relatively recent. The transductive fine-tuning method is similar to the approach proposed by Chen et al 2019 and our Baseline (finetune). Their method achieves very similar performance as the standard fine-tuning approach - a trend that can be observed in their results on standard 5-shot 5-way tasks and across different backbone architectures. None-the-less, we are willing to include more methods and results in the paper if the reviewer thinks this is the case.
> 2.  “[...] ‘backbones of different depths’ is commonly used in few-shot learning literature. Therefore it cannot reflect much contribution of this paper”
>    * A. Most of the previous work, such as Prototypical Network and MAML, originally reported results using a single backbone model (Conv4). The use of different backbones became more common only recently - an observation that was also remarked in [1] (see the first point to question 1 in appendix D of their paper).
>    * B. Note that, the use of different backbones was never intended to be a main contribution of our paper. We wanted to highlight how thoroughly we performed the comparisons - along different axes including backbones. We are open to improving the text if the reviewer thinks this is unclear.
> 3. “Some related work is not discussed [2,3].”
>    * A. We thank the reviewer for pointing out this recent work. In [2] the problem tackled appears significantly different. In particular, [2] does not consider imbalanced tasks at test-time - a key difference with our work. Moreover, the paper seems to explore meta-training on tasks with specific, constant, k(-shot) and then evaluate on tasks with a different k (e.g. meta-training on 5-shot tasks but evaluating on 1-shot tasks, etc). This work is more similar to task-distribution imbalance (Lee et al., 2020). This work is related as far as few-shot learning is but it does not consider class imbalance. We have updated the related work section (Section 2.3).
>    * B. [3] is included in our literature review, and it has been discussed. Specifically, “Chen et al., 2020 explore a pure class-imbalance problem on the support set, but their analysis is limited to just two methods (their proposal and MAML)”. The reason why it has not been implemented by us is that it offers a minimal performance advantage over MAML (according to Table 4 in their paper) but at a higher cost in terms of model complexity and implementation overhead. Bayesian TAML (Lee et at., 2020) seemed to us a more effective approach and was prioritized. None-the-less, we are willing to incorporate more baselines if the reviewer still thinks this is necessary.
> 4. “[...] more informative observations and conclusions should be made”
>    * A. We thank the reviewer for the suggestion. We have now updated the discussion section (Section 5) to further highlight novelty, discuss more insights, and in more depth, as well as offer some open questions to the community.
>
>
> [References]
> Triantafillou et al., 2020, Meta-Dataset: A Dataset of Datasets for Learning to Learn from Few Examples} ICLR 2020
> Lee et al., 2019, Learning to Balance:  Bayesian Meta-Learning for Imbalanced and Out-of-distribution Tasks, ICML 2019

---

### Official Review · AnonReviewer2 · 2020-10-28
**This paper systematically investigated the class imabalance problem in few-shot learning from multiple aspects.**

**Rating:** 5
**Confidence:** 3

**Review:**

The authors present a detailed study of few-shot class-imbalance along three axes: dataset vs. support set imbalance, effect of different imbalance distributions (linear, step, random), and effect of rebalancing techniques. The authors extensively compare over 10 state-of-the-art few-shot learning methods using backbones of different depths on multiple datasets. The analysis reveals that 1) compared to the balanced task, the performances of their class-imbalance counterparts always drop, by up to 18.0% for optimization-based methods, although feature-transfer and metric-based methods generally suffer less, 2) strategies used to mitigate imbalance in supervised learning can be adapted to the few-shot case resulting in better performances, 3) the effects of imbalance at the dataset level are less significant than the effects at the support set level.


Pros:
1) the paper covers the state-of-the-art few-shot learning methods, over 10 methods are compared in the paper;
2) the work reveals some interesting insights in few-shot learning, such as the three analysis summarized in Abstract.
3) the experiments are reasonable. There are a number of comparisons between different methods on different data sets. The codes to reproduce the experiments is released under an open-source license.

Cons:
1) the paper does not provide a new model and the contribution is marginal.
2) the experiments does not introduce new datasets as benchmark, all the datasets are heavily manipulated during testing. Is there any new data sets provides to test the assumptions of class-imbalance few-shot learning?
3) the paper does not fully discuss new possible research directions in the field of class imbalance few learning. Although the authors discuss some insight into the previously unaddressed CI problem in the (meta-) training dataset and conclude that the effects of imbalance at the dataset level are less significant than the effects at the support set level, the future work along this direction seems still unclear.

---

> ### Author Response · Authors · 2020-11-17
> **Author Response 1 to AnonReviewer2**
>
> We thank the reviewer for their positive feedback, and answer your specific comments below:
> 1. “The paper does not provide a new model and the contribution is marginal.”
>    * A. We have provided a comment on novelty in the general comment above and in the answer to AnonReviewer3. While we agree a novel method could make our contribution stronger, our paper still offers new, interesting, and impactful knowledge that brings value to the research community. To make this clearer we have updated the paper, and the discussion section (Section 5).
> 2. “the experiments does not introduce new datasets as benchmark and all the datasets are heavily manipulated during testing. Is there any new data sets provides to test the assumptions of class-imbalance few-shot learning?”
>    * A. Mini-ImageNet is a standard FSL benchmarking dataset and can be easily downloaded from the internet e.g [1][2][3]. Our specific data provider is in our source code in the supplementary material, ready for anyone to try and run. Our flexible framework allows anyone to implement their own model, balancing strategy, or even a task.
>    * B. While there are some imbalanced meta-datasets available, we were unable to find one that offered a natural imbalance at the task scale. Artificially inducing imbalance into the FSL tasks and meta-datasets allows us to precisely control it and look at the problem from multiple angles. By controlling it, we can easily compare it to the balanced tasks/datasets while maintaining a fixed support set and dataset size, and thus isolate effects of imbalance as best as possible - something that previous work has not addressed very well.
> 3. “the paper does not fully discuss new possible research directions in the field of class imbalance few learning.”
>    * A. We thank the reviewer for their suggestion and we have updated the discussion and the conclusion sections and elaborated on open questions for the community.
>
> [References]
> [1] https://github.com/yaoyao-liu/mini-imagenet-tools
> [2] https://github.com/renmengye/few-shot-ssl-public
> [3] https://www.kaggle.com/whitemoon/miniimagenet

---

### Official Review · AnonReviewer1 · 2020-10-28
**Official Blind Review #1**

**Rating:** 4
**Confidence:** 5

**Review:**

Summary

This paper introduces a new benchmark for imbalanced few-shot learning where the number of samples per class is different. The authors extensively evaluate 10 SOTA few-shot methods on this benchmark and show consistent performance drop in this challenging setting. They also show that simple over-sampling techniques can alleviate the imbalanced issue in few-shot learning.

Pros

-This paper introduces a new benchmark to evaluate the imbalance problem in few-shot learning.
-The evaluation is quite extensive and includes 10 SOTA method under different imbalanced settings

Cons

-The authors ignore the long-tail recognition literature [1, 2, 3] which is highly relevant and in my opinion, more important than the proposed imbalanced few-shot learning problem. [1] introduced a long-tail recognition benchmark where the ImageNet classes are divided into many-shot, medium-shot and few-shot classes based on the number of training examples. This setting is more realistic because the statistics of real-world datasets also follow a long-tail. I do agree that the imbalanced problem is very important. But I am not convinced that the proposed imbalanced few-shot learning (or meta-learning) evaluation protocol is appealing to the imbalanced problem community.

[1] Large-Scale Long-Tailed Recognition in an Open World. Liu et al., CVPR'19

[2] Learning imbalanced datasets with label-distribution-aware margin loss. Cao et al., NeurIPS'19

[3] Decoupling representation and classifier for long-tailed recognition. Kang et al., ICLR'20

-The authors also ignore the generalized few-shot learning literature [4, 5, 5] which is closely related to imbalanced problems and few-shot learning. In particular, [4] introduced a benchmark that evaluates the performance on both base and novel classes where base classes have many samples and novel classes have only few shot examples.

[4] Low-shot visual recognition by shrinking and hallucinating features. Hariharan et al., ICCV'18

[5] Low-shot learning with imprinted weights. Qi et al., CVPR'18

[6] Low-shot learning from imaginary data. Wang et al., CVPR'18

-There is no novelty except the proposed benchmark. The over-sampling techniques are standard and expected to improve the performance.

Justification of the rating

As an evaluation paper, the authors ignore a large group of highly relevant works in long-tail recognition and generalized few-shot learning. I think the proposed benchmark is somewhat incremental to the existing long-tail recognition benchmark and recommand a rejection.

----------------------------------------------------
Post-rebuttal

I have read the rebuttal and other reviews. The first version of this submission fails to cite any highly relevant works in long-tailed recognition and generalized few-shot learning. I believe that addressing this issue would require a major revision. The authors argue that their proposed few-shot imbalanced setting is different from the long-tailed recognition problem and generalized few-shot learning setting.  But this is only partially true. It is unclear whether previous approaches for long-tailed and generalized few-shot learning can be already applied to address the few-shot imbalanced problem. Moreover, I am not convinced that the proposed setting is more appealing than those two existing imbalanced settings because the 5-way classification problem seems to be an artificial setting that rarely happens in practice. Finally, I realize that the proposed setting is actually not new. [Lee et al., ICLR 2020] have explored a very similar setting. Thus, I would keep my original review and recommend rejection.

---

> ### Author Response · Authors · 2020-11-17
> **Author Response 1 to AnonReviewer1**
>
> [Updated sentence in 1.A for clarity]
>
> We thank the reviewer for their comments. Below we respond to the reviewer’s feedback:
> 1. “The authors ignore the long-tail recognition literature [1, 2, 3] which is highly relevant and in my opinion, more important”.
>    * A. We agree with the reviewer on the importance of studying the long-tail distribution. However, from a practical point of view it is not trivial to apply the methodology of standard long-tail imbalance in the context of few-shot learning. Few-shot learning by definition works at the ‘tail-end‘ of the long-tail distribution and does not consider classes with a large number of samples. The literature on long-tail distribution, such as [1,2,3], deals with large-scale datasets having a large amount of samples and classes. Given the differences between the two domains, a straightforward adaptation is not always possible. We have updated the paper to reflect these considerations, see related work (Section 2.1) and discussion (Section 5).
>    * B. In our opinion, the closest resemblance to the long-tail imbalance in the few-shot setting is connected with the step imbalance condition. For instance, the extreme step imbalance considered in our experiments resembles ‘long-tail‘ / ‘Pareto Law’ imbalance [1] albeit at a small-scale (see results for 1-9shot and 1-21shot step imbalance with 4-minority classes in Figure 13 in Appendix E). We can easily expand on these experiments and incorporate more settings into the paper if the reviewer thinks this is appropriate.
> 2. “I am not convinced that the proposed imbalanced few-shot learning (or meta-learning) evaluation protocol is appealing to the imbalanced problem community”
>    * A. We highlight that our work is primarily focused on the few-shot learning/meta-learning community and only tangentially to the general class-imbalance community. Our work for the first time thoroughly investigates many few-shot algorithms on the underexplored, practical, class-imbalanced task. Our review reveals new insights into the robustness and practicality of FSL algorithms, so they will mainly benefit the few-shot/meta- learning communities.
> 3. “The authors also ignore the generalized few-shot learning literature [4, 5, 5]”
>    * A. [4], [5] and [6] address a related but different problem. This work is more similar to “incremental few-shot learning” (Ren et al, 2019, Gidaris et al., 2018), where the goal is to maintain performance on the base (well-known) classes while incrementally learning about novel classes using limited data, and typically without having to re-train on the base classes. We agree that this is an interesting area of research but it has different applications from the ‘pure’ few-shot learning problem investigated in our report. Our work focuses on studying how imbalance impacts learning over novel classes only. Crucially, classical few-shot learning algorithms (eg. ProtoNet, MAML) were not initially designed to cope with incremental FSL. Extending these algorithms to the Incremental FSL could be possible (Ren et al, 2019) but it’s often not straightforward and probably merits another publication.
> 4. “There is no novelty except the proposed benchmark.”
>    * A. We have commented on novelty in the general answer above and in the answer to AnonReviewer3. We note again that the novelty and contribution is the benchmark itself. It reveals novel insights about the behaviors of FSL algorithms on imbalanced tasks, quantifies how problematic imbalance is in FSL, and offers novel insights into pairing FSL with standard rebalancing techniques, and much more. We have improved the manuscript to make this clear by adding a discussion in Section 5.
> 5. “The over-sampling techniques are standard and expected to improve the performance.”
>    * A. We agree that ROS is a standard technique and that improvement is expected; however, we noted that not all algorithms benefit from ROS equally well, and some not at all. Moreover, ROS does not entirely solve the imbalance problem. Therefore, our review encourages others in the community to develop more robust algorithms to focus on the more realistic, imbalanced task.
>
> We hope that our responses clarify our design choices and target audience. We are happy to answer any further questions/concerns and include additional results in the paper.
>
>
> [References]
>
> Ren et al, 2019, Incremental Few-Shot Learning with Attention Attractor Networks, NeurIPS 2019
>
> Gidaris et al., 2018, Dynamic Few-Shot Visual Learning without Forgetting, CVPR 2018

---

> > ### Author Response · Authors · 2020-11-19
> > **Author Response 2 to AnonReviewer1**
> >
> > In response to the concerns of the reviewer and recommendations of AnonReviewer3, we run some experiments using CB loss (Cui et al., 2019) from the long-tailed literature where we thought a straightforward adaptation was possible (see "Author Response 2 to AnonReviewer3"). However, we found no advantage over a simple Weighted Loss on the few-shot imbalance problem. These experiments suggest that the long-tailed mechanisms do not necessarily generalize to the few-shot imbalance problem.
> >
> > Cui, Yin et al. “Class-Balanced Loss Based on Effective Number of Samples.” CVPR 2019

---

### Official Review · AnonReviewer3 · 2020-10-28
**An extensive collection of results on imbalanced FSL**

**Rating:** 5
**Confidence:** 3

**Review:**

The paper analyses the effect of class imbalance on few-shot learning problems. It draws a number of interesting (but kind of expected) conclusions e.g., the support set imbalance has a larger influence on the FSL performance compared to base class imbalance, a high impact of imbalance on gradient-based meta-learning methods compared to metric learning approaches. The paper is overall

Pros:

+ The paper is nicely written with a clear structure and exposition of ideas.
+ An extensive number of FSL methods have been tested with three imbalance settings (linear, step, and random imbalance) on multiple datasets (Meta-Dataset and Mini-ImageNet) across various backbones.
+ The paper considers class imbalance in both the base training and finetuning on the support set.
+ Overall, the paper presents a thorough and detailed analysis of the class imbalance problem in FSL.

Cons:
- An approach to deal with the imbalance in FSL settings could have made the paper even more stronger.
- Specifically, two very simple rebalancing methods are studied in the paper i.e., Random over-sampling and Random shot meta-training. An algorithmic approach for appropriate rebalancing in the loss function (e.g., [a,b,c]) would be intreresting to analyze.

[a] Ren et al., Learning to Reweight Examples for Robust Deep Learning

[b] Khan et al., Cost-sensitive learning of deep feature representations from imbalanced data

[c] Cui et al., Class-Balanced Loss Based on Effective Number of Samples

---

> ### Author Response · Authors · 2020-11-17
> **Author Response 1 to AnonReviewer3**
>
> [Updated References]
>
> We thank the reviewer for the positive feedback and suggestions. Below we respond to the reviewer’s comments:
> 1. Lack of a novel approach.
>    * A. While we agree a novel method could make our contribution stronger, in our opinion we present “new, interesting, and impactful knowledge” [ICRL2021 Reviewer Guidelines] that bring value to the ICLR community. Specifically, our paper:
>       1. is the first to quantify the class imbalance problem within FSL and provide a thorough experimental review looking at the problem from multiple angles.
>       2. Is the first to expose the most and the least robust algorithms and algorithm groups (eg. optimization-based vs. metric-based) related to few-shot imbalance.
>       3. Is the first to evaluate the effectiveness of Random-Shot meta-training and show that it rarely works on its own, contrary to popular belief (Guan et al., 2020; Triantafillou et al., 2020; Lee et al., 2019; Chen et al., 2020).
>       4. is the first to pair FSL methods with simple rebalancing strategies from general class imbalance literature, such as ROS, ROS+ (with augmentation), weighted loss, focal loss
>       5. Is the first to offer insight into imbalance at the meta-training dataset level
>    * B. We argue that these insights are as important to the community as ‘yet’ another algorithm as they offer answers to many practical questions concerning the few-shot/meta-learning literature. We have updated the manuscript to make this clear (see Section 5).
> 2. “two very simple rebalancing methods are studied in the paper”
>    * A. We agree that ROS and Random-Shot meta-training are simple methods. However, their simplicity makes them very versatile. Any FSL algorithm can use ROS and it can be easily adapted to many other methods; to us this looks like a simple yet effective baseline.
> 3. “An algorithmic approach for appropriate rebalancing in the loss function (e.g., [a,b,c]) would be intreresting to analyze”
>    * A. We agree that including more rebalancing strategies could make our contribution stronger. We therefore include Figure 6 in the main paper body with the corresponding per-model results in Figure 11 in Appendix D.1.
>    * B. We thank the reviewer for pointing out the interesting work in [a] [b] [c]. We are in the process of getting the results for [c] before the end of the rebuttal period. However, we have found that [a][b] are particularly complicated to implement given the short time available for the rebuttal. This is due to peculiar technical difficulties associated with the few-shot methods we are using. For instance, it is unclear how to adapt [a] [b] to methods like MAML that requires estimation of second-order derivatives, or methods like DKT that are using Bayesian objective functions. For the moment, we have included results for weighted loss (inverse class frequency) and focal loss applied at inference-time that do not interfere with the meta-learning objective. We have found weighted loss and focal loss to be easier to implement, since there is a large amount of code that can be adapted. We think that these new results will satisfy in part the request of the reviewer and bring additional values to the paper.

---

> > ### Author Response · Authors · 2020-11-19
> > **Author Response 2 to AnonReviewer3**
> >
> > As a follow-up on response 3.B, we now have results for CB Loss ($\beta=0.8$) [c] experiments, which we can include in the paper if the reviewers think this is appropriate. Specifically, a new Figure 6 would portray a graph of the following values:
> >
> > | Imbalance, $\rho$         |       |9.00|       |4.00|       |    2.33|       |   1.50|      |   1.00 |
> > |:----------------------------|-|--------:|-|--------:|-|--------:|-|--------:|-|-------:|
> > | Standard (ROS+)             | |  52.62% | |  54.02% | |  55.11% | |  55.74% | |  56.03% |
> > | Standard (Focal Loss)       | |  34.62% | |  37.03% | |  41.00% | |  44.54% | |  50.00% |
> > | Standard (Weighted Loss     | |  52.36% | |  54.99% | |  58.97% | |  60.64% | |  61.79% |
> > | Standard (CB Loss)          | |  45.60% | |  48.90% | |  55.02% | |  58.61% | |  61.77% |
> > | Random-Shot (ROS+)          | |  53.85% | |  55.41% | |  56.45% | |  57.18% | |  58.11% |
> > | Random-Shot (Focal Loss     | |  40.24% | |  41.99% | |  46.10% | |  48.27% | |  49.96% |
> > | Random-Shot (Weighted Loss  | |  48.93% | |  51.04% | |  54.16% | |  55.40% | |  56.32% |
> > | Random-Shot (CB Loss)       | |  47.60% | |  49.68% | |  53.34% | |  55.01% | |  56.32% |
> >
> > A new per-model model graph would also replace the current Figure 11 to include the CB loss.
> >
> > We selected $\beta$ based on the $\beta=(N-1)/N$ formula as recommended in [c] where $N$ is the number of unique classes/prototypes. We also tried $\beta=(0.1,0.2,0.3,0.4,0.5,0.6,0.7,0.8,0.9,0.99)$ and found that CB Loss generally performs worse than Weighted Loss. When $\beta\approx1$, the CB Loss performs the same as Weighted Loss. This is consistent with observation and theory in [c]. These experiments suggest that the long-tailed methods do not necessarily generalize to the few-shot imbalance problem.
> >
> > [c] Cui et al., Class-Balanced Loss Based on Effective Number of Samples

---

### Author Response · Authors · 2020-11-17
**[Official Comment - To All Reviewers]**

We thank all the reviewers for their valuable feedback and time and for acknowledging that the paper is generally well written and presents a detailed analysis of the class imbalance problem in FSL. We have incorporated the following changes (all major changes are highlighted in red in the new version of the paper):
[Major]:
1. Novelty:   Some reviewers pointed out an issue with novelty since the paper does not present any new approach. In our opinion, although we do not propose a new approach, we present “new, interesting, and impactful knowledge” [ICRL2021 Reviewer Guidelines] over the few-shot class imbalance problem that has been scarcely considered so far. We think that our findings can bring value to the research community, helping in developing new methods and have an impact in many real-world applications. Note that, our review is similar in spirit to other review papers accepted at previous editions of ICRL (e.g. Chen et al., 2019). To improve this point, we have now added a discussion (Section 5) where we give an overview and discuss the key results of the paper.
2. Related Work: We have expanded the related work section (Section 2) to include literature on long-tail class imbalance and some FSL and CI work, as requested by AnonReviewer1 and AnonReviewer4.
3. Additional rebalancing methods: We have included additional results using two rebalancing cost functions, Weighted Loss and Focal Loss (Figure 6, and Figure 11 in Appendix D.1) in the main body of text, as requested by AnonReviewer3. We note that Figure 11 now replaces the old Figure 10 that contained now outdated focal-loss results.

[Minor]:
* We corrected a couple of typos in the main body of the paper and a few throughout the Appendix.
* We corrected the lettering of a couple of appendices that got mixed up before submission.
* We rephrased a couple of sentences to allow for more efficient spacing.


We believe your valuable comments and our latest changes make the current revision much stronger, more informative, and valuable for the research community.


Chen et al., 2019, A Closer Look at Few-shot Classification, ICLR2019,

---

### Decision · Program_Chairs · 2021-01-07
**Final Decision**

**Decision:**

Reject

**Comment:**

The paper studies the effectiveness of few-shot learning techniques in settings where the training labels are imbalanced. While addressing an interesting practical problem, reviewers raised concerns about the paper's technical depth, insufficient distinction to existing techniques for coping with label imbalance, and limited qualitative conclusions from the results. The authors incorporated some of these comments in their revision, but a more comprehensive update on the latter two points appears appropriate.